# Cell-mediated cytotoxicity within CSF and brain parenchyma in spinal muscular atrophy unaltered by nusinersen treatment

I-Na Lu [1], Phyllis Fung-Yi Cheung[2,3,4], Michael Heming [1], Christian Thomas [5], Giovanni Giglio [2,3,4], Markus Leo[6], Merve Erdemir[6], Timo Wirth [1], Simone König[7], Christine A. Dambietz[1], Christina B. Schroeter [8], Christopher Nelke[8], Jens T. Siveke [2,3,4], Tobias Ruck [8], Luisa Klotz [1], Carmen Haider[9,10], Romana Höftberger[9,10], Christoph Kleinschnitz[6], Heinz Wiendl [1], Tim Hagenacker [6,11] ✉ & Gerd Meyer zu Horste [1,11] ✉

5q-associated spinal muscular atrophy (SMA) is a motoneuron disease caused by mutations in the survival motor neuron 1 (SMN1) gene. Adaptive immunity may contribute to SMA as described in other motoneuron diseases, yet mechanisms remain elusive. Nusinersen, an antisense treatment, enhances SMN2 expression, benefiting SMA patients. Here we have longitudinally investigated SMA and nusinersen effects on local immune responses in the cerebrospinal fluid (CSF) - a surrogate of central nervous system parenchyma. Single-cell transcriptomics (SMA: N = 9 versus Control: N = 9) reveal NK cell and CD8+ T cell expansions in untreated SMA CSF, exhibiting activation and degranulation markers. Spatial transcriptomics coupled with multiplex immunohistochemistry elucidate cytotoxicity near chromatolytic motoneurons (N = 4). Post-nusinersen treatment, CSF shows unaltered protein/transcriptional profiles. These findings underscore cytotoxicity's role in SMA pathogenesis and propose it as a therapeutic target. Our study illuminates cell-mediated cytotoxicity as shared features across motoneuron diseases, suggesting broader implications.

5q-associated spinal muscular atrophy (SMA) is an autosomal-recessive degenerative neuromuscular disorder associated with progressive loss of motoneurons and clinically characterized by a progressive muscle weakness with severe impairment of motor function. Approximately 94% of SMA patients carry a homozygous exon 7 deletion of the Survival Motor Neuron 1 gene (*SMN1*), that abrogates expression of the SMN protein[1,2]. SMN serves a predominant function in spinal motoneurons, and its complete deficiency causes early lethality and a severe clinical picture of SMA[3]. Patients survive due to low amounts of the SMN protein being translated from transcripts of the

[1]Department of Neurology with Institute of Translational Neurology, University Hospital Münster, Münster, Germany. [2]Spatiotemporal Tumor Heterogeneity, German Cancer Consortium (DKTK), Partner Site Essen, A Partnership Between German Cancer Research Center (DKFZ) and University Hospital Essen, Essen, Germany. [3]Bridge Institute of Experimental Tumor Therapy, West German Cancer Center, University Hospital Essen, University of Duisburg-Essen, Essen, Germany. [4]Division of Solid Tumor Translational Oncology, DKTK, Partner Site Essen, A Partnership Between German Cancer Research Center (DKFZ) and University Hospital Essen, Essen, Germany. [5]Institute of Neuropathology, University Hospital Münster, Münster, Germany. [6]Department of Neurology and Center for Translational Neuro and Behavioral Science, University Hospital Essen, Essen, Germany. [7]Core Unit Proteomics, Interdisciplinary Center for Clinical Research, University of Münster, Münster, Germany. [8]Department of Neurology, Medical Faculty and University Hospital Düsseldorf, Heinrich Heine University Düsseldorf, Düsseldorf, Germany. [9]Division of Neuropathology and Neurochemistry, Medical University of Vienna, Vienna, Austria. [10]Comprehensive Center for Clinical Neurosciences and Mental Health, Medical University of Vienna, Vienna, Austria. [11]These authors contributed equally: Tim Hagenacker, Gerd Meyer zu Horste. ✉e-mail: tim.hagenacker@uk-essen.de; gerd.meyerzuhoerste@ukmuenster.de

orthologous *SMN2* gene, a nearly identical copy of *SMN1*. Hence, the SMA disease severity is mainly determined by the level of compensatory transcription of available *SMN2* copies and is clinically classified by the highest motor milestone achieved and disease-onset as type I (infantile-onset, sitting is not achieved), type II (late-onset, walking is not achieved), and type III (late-onset, walking is achieved)[4].

Previous studies in SMA animal models demonstrated that loss of the SMN protein leads to an activation of glia cells, such as astrocytes and microglia, and resulted in a higher expression of pro-inflammatory and pro-apoptotic cytokines such as TNFα, IL-1 and IL-6 as well as complement factors such as C1q and C3[5–11]. This suggested an involvement of immunological mechanisms in SMA. In type II and type III SMA patients, central nervous system (CNS) observations unveiled inflammatory-associated alterations, particularly marked by gliosis[12,13]. Recent findings also revealed an impairment of lymphoid organ functions in SMA, which may mediate neurodegenerative processes through inflammation and by contributing to the pathogenesis of SMA[14–16]. In a murine model of another motoneuron disorder, familial amyotrophic lateral sclerosis (ALS), cytotoxic lymphocytes such as CD8+ T cells and NK cells were shown to trigger death of spinal motoneurons[17,18]. Signs of cytotoxicity were also reported in the cerebrospinal fluid (CSF) and brains of some ALS patients[19,20]. Whether such cytotoxicity also occurs in SMA – a monogenetically defined and per se treatable motoneuron disease – remains unknown. It is worth noting, however, that SMA is typically considered a non-cell autonomous disease, where inflammatory changes appear to precede motor neuron loss[21].

The first approved drug for the treatment of SMA named nusinersen is an antisense oligonucleotide, which is administered intrathecally into the CSF with a loading phase in the first weeks and ongoing treatment three times per year[2,22]. Nusinersen induces alternative splicing of the SMN2 gene, functionally converting it into SMN1 transcript and thus increasing the level of SMN protein in the CNS and ameliorating disease severity[23]. Considering that SMN2 is also expressed in immune cells[17], and that nusinersen is applied directly into the CSF which is populated by immune cells, one could speculate that nusinersen has the potential to directly modulate immune cells in the CSF. A recent study in pediatric SMA patients observed that repeated intrathecal injections of nusinersen did not trigger unwanted inflammatory responses. Instead, it revealed an increase in a neuroprotective protein, monocyte chemoactive protein 1 (MCP1/CCL2), during the course of nusinersen treatment[24]. SMA patients also exhibit an inflammatory profile in both CSF[25–27] and serum[26]. Additionally, variations in specific cytokine levels were found, with severe type I SMA patients showing higher pro-inflammatory cytokine levels in their CSF compared to milder type II and type III cases. Nusinersen had differing effects on these cytokines in SMA type I, type II and type III patients[25]. These studies also hinted at potential connections between specific cytokines and motor function outcomes, emphasizing the intricate role of neuroinflammation and the immune system in SMA[24–26].

Single-cell transcriptomics studies of the CSF have recently advanced the pathogenetic understanding of neurological diseases by providing unprecedented resolution[28,29]. Here we applied single-cell RNA sequencing (scRNA-seq) to longitudinal patient material to first generate a comprehensive transcriptional map of CSF immune cells in SMA patients before, at 6 months, and at 10 months under nusinersen treatment. We identified an expansion of cell-mediated cytotoxicity in the CSF of untreated SMA patients and found that these cytotoxic NK/CD8+ T cells were likely recruited and activated by IL-18- and CCL5-secreting monocyte subsets. This observation was further supported by CSF proteomics. Nusinersen treatment did not modulate these abnormalities. Spatial transcriptomic and multiplex immunohistochemistry detected cytotoxic cells in the vicinity of degenerative neurons (DeN) as well as IL-18 expressing monocytes in the brain of

type 1 SMA patients indicating that macrophage-induced recruitment of cytotoxic cells may contribute to SMA. The concept of cytotoxicity promoting loss of motoneurons[18,30,31] may thus be generalizable across motoneuron diseases with potential for therapeutic targeting in SMA and beyond.

## Results

### Intrathecal treatment in SMA provides unique access to longitudinally collected CSF

We sought to understand how the monogenetically defined motor neuron disease SMA shaped cellular immunity in the vicinity of the brain. We capitalized on the fact that nusinersen treatment for SMA requires repeated intra-CSF (i.e., intrathecal) application[23,32] thus providing unique access to longitudinally gathered CSF. We collected CSF from SMA patients immediately before nusinersen application (SMA_baseline) and after 2, 6 and 10 months of ongoing treatment. The clinical characteristics of the patients are provided in Table 1. CSF cells from Cohort 1 were analyzed by scRNA-seq, while cell-free CSF supernatants from Cohort 2 were examined by high-definition mass spectrometry (HDMS)- and Olink-based proteomics.

In Cohort 1 we analyzed all cells in freshly collected CSF samples by scRNA-seq. After excluding samples with low volume or blood contamination (see "Methods"), we successfully processed 23 individual CSF cell samples from 7 individual SMA patients that were longitudinal (baseline: $n = 7$; 2mo: $n = 2$; 6mo: $n = 6$; 10mo: $n = 6$) and 2 additional SMA patients at 10 months post treatment without available baseline data (10mo: $n = 2$; Supplementary Data 1).

We integrated these data with available data from 9 control patients[33] with idiopathic intracranial hypertension (IIH); a non-inflammatory neurological disease featuring excessive CSF pressure. In total, this resulted in 88,672 single-cell transcriptomes (2860 mean ± 382 SD cells per sample) in the entire dataset (1483 mean ± 134 SD genes detected per cell). A total of 22,303 single-cell transcriptomes (henceforth named cells for simplicity) were available from controls and 66,369 cells from all SMA patients.

We then performed clustering and annotated the resulting 15 clusters based on the expression of predefined marker genes (Fig. 1A, B; Fig. S1A, B; Supplementary Data 2). We identified a monocyte/granulocyte cluster (Mono1: *CD14, LYZ, S100A8, S100A9*), and two CSF macrophages clusters, which resembled border-associated macrophages (CSF_M1, CSF_M2, *MRC1, LYVE1, CD14*) as previously described[34]. Three clusters expressed DC markers (cDC1: *CLEC9A, XCR1*, cDC2: *CLEC10A, CD1C, FCER1A*, pDC: *CLEC4A, TNFRSF21, IRF8*). Five clusters expressed T cell markers (CD4_1–3 and CD8_1–2: *CD3E, TRAC, CD40LG*), among them CD4_1 showed naïve CD4+ T cell features, including *CCR7, MAL, IL7R, MOSIP*, and *LDHB*. CD4_2 exhibited markers of effector/memory CD4+ T cells, such as *IL2RG, KLRB1, IL32*, and *TRAC*; while CD4_3 displayed NKT cell features, like *NKTR, C1orf56*, and *KLRK1*. In addition, we identified one of the two CD8+ T cell clusters, namely CD8_2, with strong induction of the *GNLY* gene indicative of cytotoxicity. Furthermore, we detected an NK cell cluster (NK: *NKG7, GNLY*) and a B cell cluster (B: *MS4A1*/CD20, *CD79B, CD24*). We found one cluster of proliferating cells (prolif: *CDCA3, CDK1, MKI67*) and one cluster of PTN+ cells (PTN+: *PTN, MDK, LYZ*) that most likely represented a specific subset of myeloid lineage cells. This replicated the known composition of CSF cells being exclusively of hematopoietic origin[30,33,35,36].

### Single-cell transcriptomics identifies an increase of cytotoxicity in the CSF of SMA patients

We next aimed to understand whether and how SMA-affected CSF cells. We first tested for differential cluster abundance (DA) between SMA_baseline and Control. We identified a significant increase of both the frequency and absolute count of CD8_2 at SMA_baseline versus controls (Fig. 1C; Fig. S2C) while there was no difference of total CSF

**Table 1 | Cohort characteristics**

| Cohort Characteristics | Cohort 1 (scRNA-seq) | | Cohort 2 (Proteomics) | |
|---|---|---|---|---|
| | IIH Control | SMA | IIH Control | SMA |
| **No. of subjects** | 9[a] | 9 | 10 | 10 |
| *By diagnosis group* | | | | |
| SMA III | – | 7 | – | 8 |
| SMA II | – | 2 | – | 1 |
| SMA I | – | 0 | – | 1 |
| **Sex (No.)** | | | | |
| Male | 2 | 7 | 6 | 6 |
| Female | 7 | 2 | 4 | 4 |
| **Age [years (mean ± SD)]** | 31.7 ± 7.5 | 32.1 ± 10.8 | 35.8 ± 13.4 | 33.7 ± 13.4 |
| **SMN2 copy number** | | | | |
| 4 (%) | – | 5 (56%) | – | 3 (30%) |
| 3 (%) | – | 4 (44%) | – | 6 (60%) |
| 2 (%) | – | 0 (0%) | – | 1 (10%) |
| **Hammersmith functional motor scale expanded (HFMSE)** | | | | |
| SMA_baseline (mean ± SD) | – | 42.7 ± 9.0 | – | 27.5 ± 26.2 |
| SMA_2mo (mean ± SD) | – | 41.0 ± 4.2 | – | – |
| SMA_6mo (mean ± SD) | – | 38.0 ± 7.7 | – | 30.0 ± 25.8 |
| SMA_10mo (mean ± SD) | – | 40.0 ± 6.6 | – | – |
| **CSF protein level in milligram per deciliter (mg/dl)** | | | | |
| SMA_baseline (mean ± SD) | – | 34.0 ± 3.8 | – | 41.6 ± 13.2 |
| SMA_2mo (mean ± SD) | – | 35.5 ± 0.7 | – | – |
| SMA_6mo (mean ± SD) | – | 41.3 ± 17.8 | – | 41.2 ± 11.5 |
| SMA_10mo (mean ± SD) | – | 44.3 ± 18.7 | – | – |
| **Presence of spondylodesis** | | | | |
| Yes (%) | – | 2 (22%) | – | 3 (30%) |
| No (%) | – | 7 (78%) | – | 7 (70%) |
| **Symptom duration in years** | | | | |
| Mean ± SD | – | 27.7 ± 13.2 | – | 32.7 ± 12.1 |
| Range | | 41 | | 39 |
| Minimum | – | 13 | – | 16 |
| Maximum | – | 54 | – | 55 |

Sample size, sex characteristics, average age, SMA diagnosis, SMN2 copy number, HFMSE score, presence of spondylodesis and symptom duration are provided for study cohorts.
[a]Previously published scRNA-seq data[33,34].

cell counts and of other cell clusters between SMA_baseline and controls (Fig. S2A). Remarkably, CD8_2, along with NK, emerges as one of the clusters expressing the highest levels of GNLY (Fig. 1B). Differentially expressed genes (DEGs) analysis identified a significant upregulation of the GNLY gene in CD8_2 compared to CD8_1 cells (Fig. S7A). Granulysin (encoded by GNLY) is a cytolytic molecule expressed by cytotoxic T cells during the advanced stages of activation[37,38]. Consistent with this observation, Gene Ontology (GO) enrichment analysis returned biological functions related to the later stages of cytotoxicity, such as "negative regulation of cell killing" and "negative regulation of leukocyte-mediated cytotoxicity" exclusively associated with CD8_2 cells (Fig. S7B).

We next tested for DEG comparing SMA_baseline versus controls and found that CD8_2 and NK clusters expressed significantly higher levels of cytotoxic genes. Specifically, in a SMA_baseline versus controls comparison, transcripts such as *CD96, NPC1, IFNG-AS1, PRF1, TNFSF12, GZMM*, and *IL17RA* were induced in the CD8_2 cluster (Fig. 1D; Supplementary Data 3) and *NCR1, NCAM1, IFNG-AS1, CD96, IL18R1, IL32*, and *GZMM* in the NK cluster (Fig. 1E; Supplementary Data 3). This suggests an increased cytotoxicity of CD8+ T cells and NK cells in SMA. Pathway enrichment analysis of DEG between SMA_baseline versus Controls within the CD8_2 and NK clusters also returned immune

activation pathways (e.g., T cell receptor (TCR) signaling pathway, MAPK cascade and Fc receptor signaling pathway) (Fig. 1F, G). SMA is thus associated with both an increased abundance and a more aggressive late activation state of cytotoxic CD8+ T cells and NK cells in the CSF.

We next asked whether the increase in cytotoxic cell types and cytotoxic phenotype was specific to the CSF compartment. We therefore performed flow cytometry of cryo-preserved peripheral blood mononuclear cells (PBMCs) of untreated SMA patients compared to sex-matched controls to examine the frequencies of cytotoxic NK and CD8 T cells and found no significant differences (Fig. S3) at low statistical power. This suggests that an expansion of cytotoxic CD8+ T and NK cells may be CSF-specific.

**Distinct signaling pathways identified in the CD8_2 and NK cells of untreated SMA patients**
Aiming to understand what drives cytotoxicity in the CSF in SMA, we next inferred cell–cell communication using CellChat[39] (Supplementary Data 4). We further compared the information flow for each signaling pathway between SMA_baseline and Control. The information flow is quantified by aggregating the probabilities of communication between all pairs of cellular groups in the inferred network. Multiple

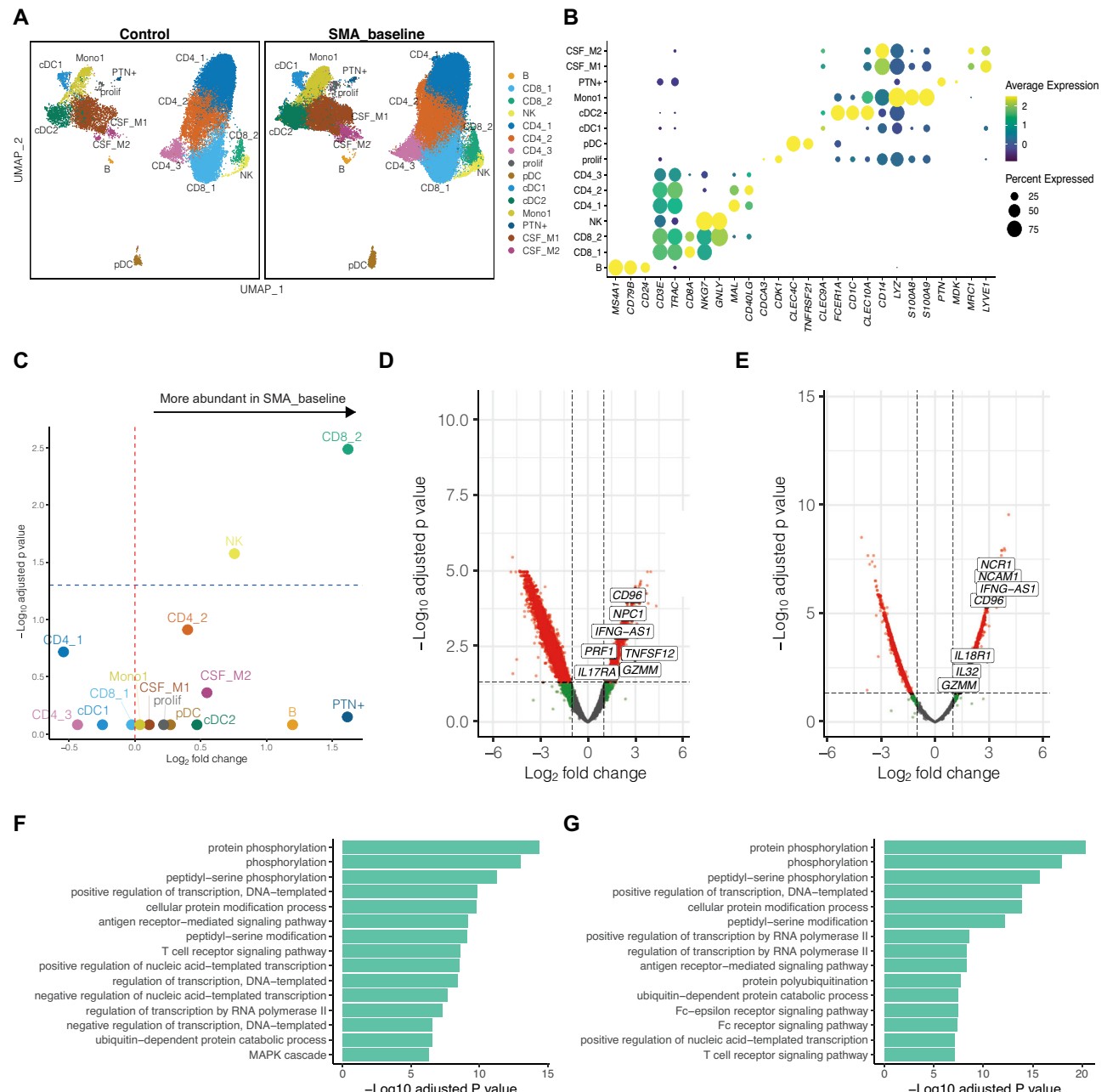

**Fig. 1 | scRNA-seq reveals an increase of cytotoxic cells in the CSF of SMA patients. A** UMAP plot showing 15 color-coded cell clusters of CSF cells from SMA_baseline patients (*N* = 7, cell count = 21,660) and Control donors (*N* = 9, cell count = 17,444). **B** Dotplot showing marker genes of cell clusters. Color encodes average gene expression; dot size represents percentage of cells expressing the gene. **C** Volcano plot depicting changes in cluster abundances in SMA_baseline (*N* = 7) versus Control (*N* = 9). Statistical significance calculated via propeller *t*-test, which uses a moderated *t*-statistic based on the robust empirical Bayes method from limma. *P* values were adjusted with the Benjamini–Hoch method. Logarithmic

fold change of cluster abundance is plotted against negative logarithmic adjusted *p* value. The horizontal lines represent the significance thresholds (blue-dashed: *p* = 0.05). **D**, **E** Differentially expressed genes (DEG) in CD8_2 (**D**) and NK (**E**) clusters in SMA_baseline (*N* = 7) versus Control (*N* = 9). Statistical significance was calculated in limma using a moderated *t*-statistic based on the robust empirical Bayes method in limma. *P* values were adjusted with the Benjamini–Hochberg method. **F**, **G** GO terms for biological processes enriched in the DEG in CD8_2 (**F**) and NK (**G**) clusters in SMA_baseline versus Control. Source data are provided as a Source Data file.

pathways were induced in SMA_baseline but not in control, including PTN, TGFb, IL16, COMPLEMENT, CCL, IL1 (Fig. 2A). The majority of these pathways are involved in inflammatory responses. Differential communication analysis further revealed that both CD8_2 and NK clusters in SMA_baseline displayed more incoming and outgoing signaling than Control (Fig. 2B, C). The top differential incoming signaling were PTN, CD70, BTLA for CD8_2 cells and PTN, TGFb, IL1 for NK cells, respectively (Fig. 2B, C). We next sought to identify the source of these

signals (Fig. 2D–I). The top differential incoming signaling, apart from TGNb, was sent by myeloid cells with antigen-presenting potential, such as PTN by PTN+ cells (Fig. 2D), IL18 by several clusters (Fig. 2H). Moreover, IL1 signaling pathway, known to be involved in cytotoxicity activation especially through IL18-IL18R1/IL18RAP ligand–receptor interaction[40], was also observed exclusively in SMA_baseline but not in Control, indicating IL-18 signaling as a potential underlying recruiting mechanism of inducing cytotoxic immune responses in SMA.

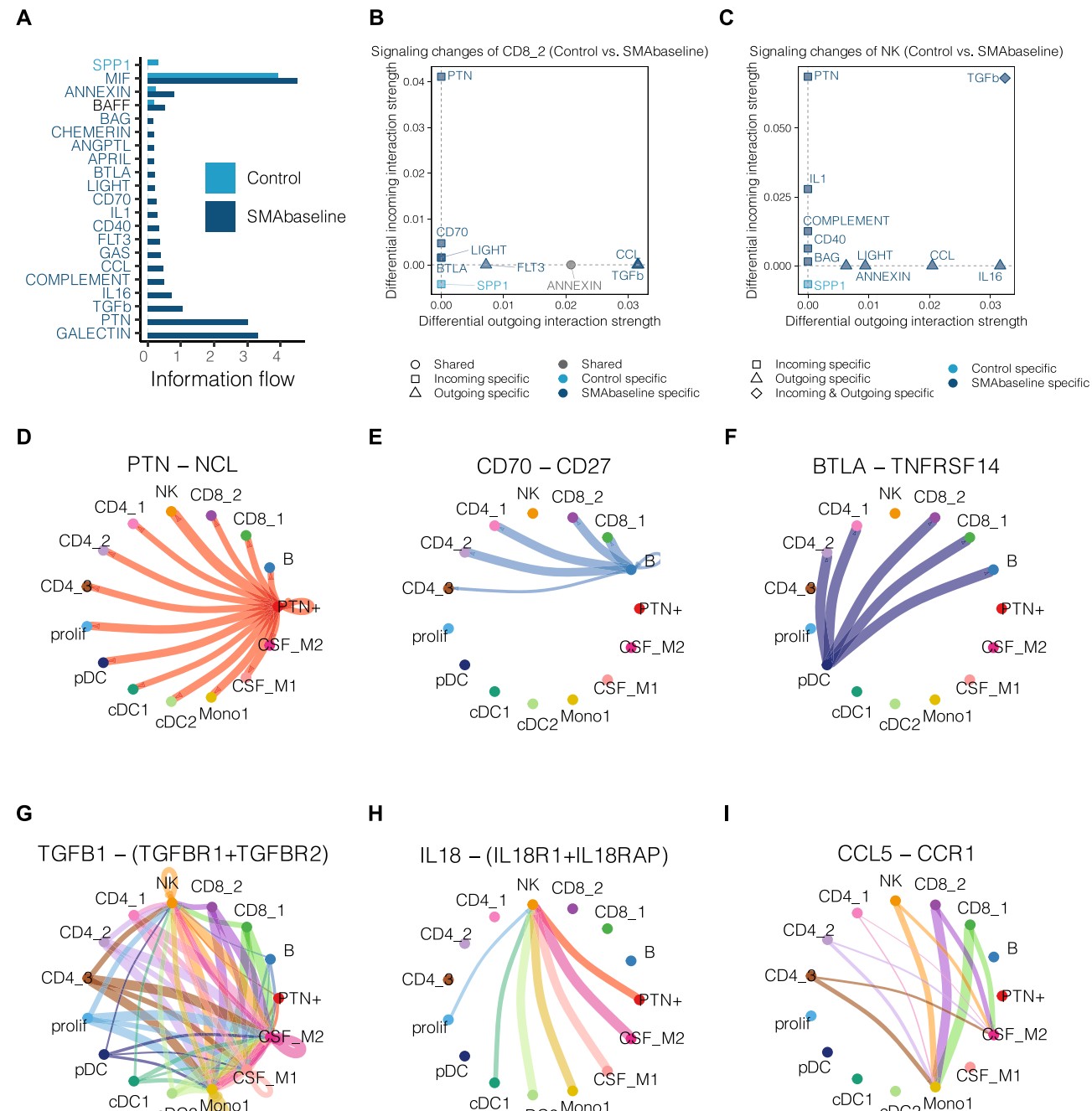

**Fig. 2 | SMA alters cell−cell interaction preferentially in cytotoxic cells.**
**A** Significant signaling pathways are ranked based on their differences of overall information flow, calculated by summarizing all communication probabilities among all pairs of cell groups using CellChat. SPP1 signaling pathway labeled in light blue is more enriched in Control. BAFF signaling pathway labeled in black is equally enriched in Control and SMA_baseline. The other signaling pathways labeled in dark blue are more enriched in SMA_baseline. **B**, **C** Altered CD8_2-associated (**B**) and NK-associated (**C**) signaling pathways in Control versus SMA_-baseline. The *x* and *y* axes indicate differential outgoing and incoming interaction strengths, respectively. Gray, light blue, and dark blue colors represent whether a signaling pathway is shared between Control and SMA_baseline, or specific to either Control or SMA_baseline. **D−I** Circle plots of the significant interactions (ligand−receptor pairs) in the selected SMA_baseline specific signaling networks from (**B**, **C**), including PTN-NCL of the PTN signaling pathway (**D**), CD70−CD27 of the CD70 signaling pathway (**E**), BTLA-TNFRSF14 of the BTLA signaling pathway (**F**), TGFB1-(TGFBR1 + TGFBR2) of the TGFb signaling pathway (**G**), IL18-(IL18R1 + IL18RAP) of the IL1 signaling pathway (**H**), and CCL5−CCR1 of the CCL signaling pathway (**I**).

## Cytotoxic lymphocytes are detected in an SMA-affected CNS region

We next examined whether cytotoxic immune responses were also present in the brain of patients with SMA. We obtained histological samples of brainstem tissue of type 1 SMA patients ($N = 4$) and first deeply characterized one of these samples. Hematoxylin and eosin

(H&E) staining suggested neuronal chromatolysis within this SMA brain tissue (Fig. S4A). We next applied an integrative single-nucleus transcriptomics strategy with single nuclei pathology sequencing (snPATHO-seq)[41] together with spatial transcriptomics. Commercially available spatial transcriptomics (Visium) was integrated with snPATHO-seq (Fig. S4B). After filtering out low-quality nuclei, we

obtained a total of 12,700 nuclei from the snPATHO-seq sample, with a median of 1329 genes detected per nucleus. The spatial transcriptomics dataset contained a total of 1431 spots and a median of 729 genes per spot. We clustered cells on the basis of the snPATHO-seq data and identified 18 brain cell subtypes (Fig. S4C; Supplementary Data 5). These cell subtypes were further categorized into eight broad brain cell type classes[42] (Fig. S4D). One cell type, annotated as DeN expressed high levels of *C1QL1* (Fig. 3A) as well as *NEFL*, *SNCA* and *TREM2* genes; transcripts indicative of neuronal damage[43–46] and microglia reactivity[47] (Fig. S4E–H). Notably, we identified genes involved in the recruitment of cytotoxic cells, including *IL18* and *CCL5*, and markers of cytotoxicity (*GZMB, GZMM*) (Fig. 3A). Remarkably, these granzyme transcripts were localized in close proximity to DeN.

We next aimed to confirm cytotoxic lymphocytes in the available SMA brain tissues (*N* = 4) on a protein level. We applied antibody staining and found CD8+ cells in the vicinity of chromatolytic motoneurons (Fig. 3B). Immunohistochemical staining for Caspase-3 suggested apoptosis in the identified chromatolytic motoneurons (Fig. 3C). We further performed multiplex immunofluorescence (IF) to distinguish CD8+ cells (CD8+) and myeloid cells (CD68+) in the SMA brains and in the control brains (Fig. 3D, E; Fig. S5; "Methods"). We next quantified GzmB+CD8+ cells and found a higher amount of GzmB +CD8+ cells in the SMA brains than in the control brains (Fig. 3F). Significantly more GzmB+CD8+ cells were detected in close proximity (≤100 μm radial distance) of IL-18+CD68+ myeloid cells in the SMA brains (Fig. 3G, H), implying that the secretion of IL-18 by myeloid cells within the CNS represents a potential recruitment/activation mechanism locally driving cytotoxic immune responses triggering neurodegeneration, which confirms the transcriptomic evidence in the CSF on a tissue-level in the brain.

### Effect of nusinersen therapy on the CSF immune landscape of SMA patients

Nusinersen therapy mainly acts by reducing loss of motoneurons, while one study also indicated immunomodulatory effects[24]. Hence, we aimed to understand whether nusinersen affected CSF leukocyte composition. We did not find significant alterations of CSF immune cell composition after 6 and 10 months of nusinersen therapy compared to baseline (Fig. S6; Supplementary Data 6).

We next reconstructed single-cell TCR sequence information (Fig. S7A) despite their limited number in CSF resulting in considerable variability in the retrievable TCR information. Fewer CSF cells were available in the 10-month-treated patient samples (Supplementary Data 1). Given this inherent limitation, performing downsampling to equalize sample sizes becomes infeasible. We compared the true diversity index and clonal proportions across samples and groups (Fig. S7B, E, F). Our findings revealed a mild predominance of larger clones (clone frequency > 5) in SMA patients, with no statistically significant alterations observed in TCR diversity or clonal proportions post-treatment or when compared to the control group (Fig. 4A; Fig. S7E, F). This indicates a non-significant trend toward increased CD8_2 clonal expansion in SMA samples (Fig. S8). Clonal expansion in SMA thus appears less pronounced than described in ALS CSF and brains[19,20] potentially due to shorter disease duration and lower patient age in type 1 SMA ("Methods"). Additionally, we have conducted public clonotype analysis and applied hierarchical clustering analysis to estimate the similarity of samples by considering the number of shared clonotypes and to assess the distance between samples (Fig. S7C, D). Our findings indicate that shared clonotypes are primarily observed within samples collected at different time points from the same patient. However, there is one intriguing exception: the clonotype with the CDR3 sequence – CSVVDTEAFF, which is shared between patients SMA1-1 and SMA1-6. We found no TCR clonotype that is shared among patients and linked to any viral infection upon analysis of public VDJ databases.

We further compared the signaling alterations of the CSF CD8_2 and NK cells before and after nusinersen treatment in the scRNA-seq data of CSF. Pro-inflammatory pathways, such as *PTN*, *CCL* and *LIGHT* were only identified in SMA_baseline and were no longer detected after treatment (Fig. 5A–D). Moreover, *TGFb* and *ANNEXIN* signaling pathways were also significantly reduced after treatment (Fig. 5A–D). These pathways are modulated by *TGFB1* and *ANXA1*, respectively, which are anti-inflammatory mediators that are crucial for the effective and selective removal of apoptotic neurons under neuroinflammatory conditions[48–50], indicating the potential regulatory mechanisms on SMA-associated neuroinflammation by nusinersen on the cellular level.

Last, we analyzed the proteomic profile of cell-free CSF supernatants of SMA patients before and after 6 months of treatment in a separate cohort (Cohort 2, Table 1). We employed two proteomic profiling methods: an unbiased HDMS-based approach (Supplementary Data 7) and a targeted inflammation panel using proximity extension assay (PEA) technology (Olink, Supplementary Data 8). Our analysis revealed a consistent trend in which cytotoxicity-associated proteins, including IL-18R1, CD5, PD-L1 and TNFRSF9, exhibited reductions after 6 months of nusinersen treatment, corroborating our transcriptome findings (Fig. S9). Additionally, we observed decreased levels of two other inflammatory proteins, CX3CL1 and MMP-1, following treatment (Fig. S9). These reductions demonstrated statistical significance via the Wilcoxon signed-rank test but did not reach statistical significance when rigorously accounting for multiple hypothesis testing[51] ("Methods"). Nusinersen thus did not induce statistically significant alterations of CSF protein composition (Fig. S9; Supplementary Data 8). Cytotoxicity in CSF was thus unaltered by nusinersen treatment.

### Discussion

Our study identified an expansion of cytotoxic NK cells and CD8+ T cells in both the CSF and brains of untreated SMA patients, with signs of activation and degranulation. This identifies cytotoxicity as a shared feature across neurodegenerative diseases. Other neurodegenerative disorders, such as Alzheimer's disease or sporadic amyotrophic lateral sclerosis (sALS), usually have mixed genetic or unknown causes. Moreover, the presence of cytotoxic lymphocytes in the brain tissues of SMA type 1 patients, in the vicinity of chromatolytic motoneurons, supports the functional role of these cells in the pathogenesis of SMA. Transcriptomics and histology suggest that cytotoxic lymphocytes may be recruited and activated by IL-18-secreting myeloid cells in the CNS and not directly by lytic neurons. How these myeloid sense neuronal destruction remains to be determined. The present findings align with prior investigations in ALS that have demonstrated the involvement of cytotoxic lymphocytes, particularly CD8+ T cells and NK cells, in the induction of motoneuron degeneration[18,31]. Recent studies on various neurodegenerative disorders have also supported this notion. Notably, a study of Alzheimer's disease detected clonally expanded CD8+ T cells within the CSF of afflicted patients, indicating that cytotoxic immune responses may contribute to the pathology of the condition[30]. Altogether, cytotoxicity might be a generalizable feature across multiple neurodegenerative diseases, which hints toward therapeutic potential.

Recent studies have investigated the immune alterations in SMA, mainly through animal research. Two major areas of immune alteration have been identified: neuroinflammation and abnormalities in lymphoid organs as a consequence of *SMN1* deficiency. Increased activation of microglia and monocytes, along with heightened production of TNF-α, NO and complement factors, have been reported in SMA animals[8,10]. In SMA△7 mouse spinal cords, activation of astrocytes and ERK1/2 phosphorylation have been observed, leading to activation of the apoptotic pathway and increased expression of pro-inflammatory cytokines[9]. Autopsies of SMA type I patients have confirmed alterations in the spleen[15]. While the precise impact of

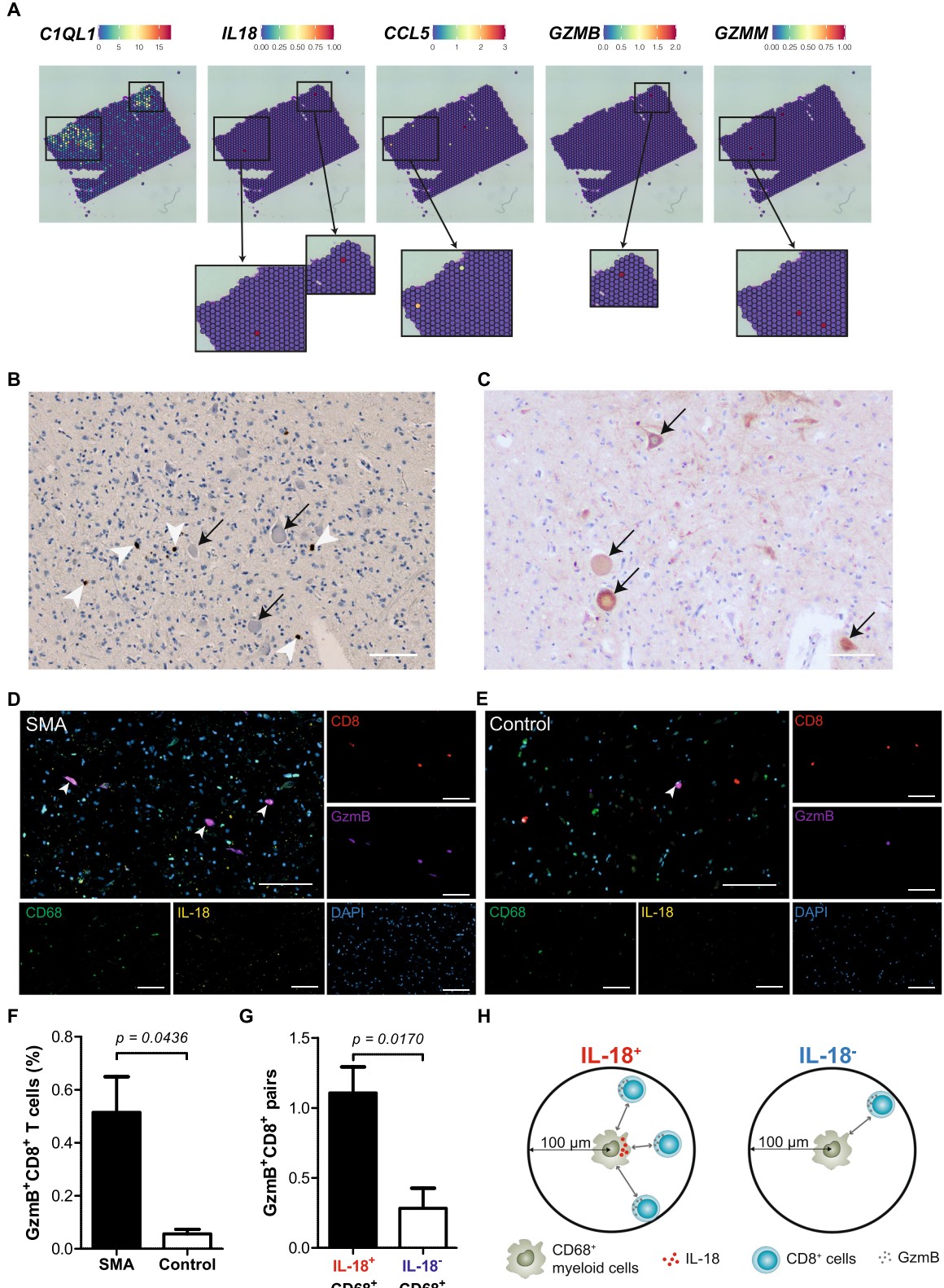

nusinersen treatment on the immune system of SMA patients remains elusive, one recent study in SMA children under 9.2 years old showed that repeated intrathecal injections are safe and do not cause unwanted inflammatory responses[24]. Additionally, the study observed an increase in the neuroprotective protein MCP1 in the CSF during nusinersen treatment, suggesting a potential neuroprotective mechanism in pediatric SMA[24]. An intriguing discovery from another study by

ref. [25] unveiled a significant reduction in IL-1ra levels following a 10-month course of nusinersen treatment in type III SMA patients. In contrast, our CSF proteomic study, focusing on adult type III SMA patients after a 6-month course of nusinersen treatment, did not reveal statistically significant changes in MCP1, IL-1ra or other major inflammatory mediators. However, our comprehensive analysis, incorporating a 92-plexed panel of inflammatory molecules, consistently

**Fig. 3 | Multimodal spatial transcriptomics of SMA brain identifies cytotoxic neuronal damage. A** Spatial GEX of *C1QL1, IL18, CCL5, GZMB* and *GZMM* genes. Each spot is 55 μm in diameter. **B** IHC staining of CD8 in SMA brains. White arrowheads indicate CD8+ cells; black arrows indicate chromatolytic neurons (*N* = 4). Scale bar: 100 μm. **C** Caspase-3 staining in SMA brains to detect apoptosis (*N* = 4). Black arrows indicate apoptotic neurons. Scale bar: 100 μm. **D**, **E** Multiplexed immunofluorescence histological staining of CD8 (red), Granzyme B (GzmB; violet), IL-18 (yellow) and CD68 (green) in brain tissue sections of SMA (**D**)

and Control (**E**). White arrowheads indicate GzmB+CD8+ cells. Scale bar: 100 μm. **F** Percentages of GzmB⁺CD8⁺ cells in brain tissue sections of SMA (*N* = 4) versus Control (*N* = 2). Mean + SEM are shown. Two-tailed Welch's *t*-test was used to calculate *p* values. **G, H** The number of GzmB⁺CD8⁺ cells in proximity (<100 μM radical distance) of IL-18⁺CD68⁺ or IL-18⁻CD68⁺ cells in the SMA samples (*N* = 4). Mean + SEM are shown. Two-tailed Welch's *t*-test was used to calculate *p* values. The diagram was created with BioRender.com and Adobe Illustrator. Source data are provided as a Source Data file.

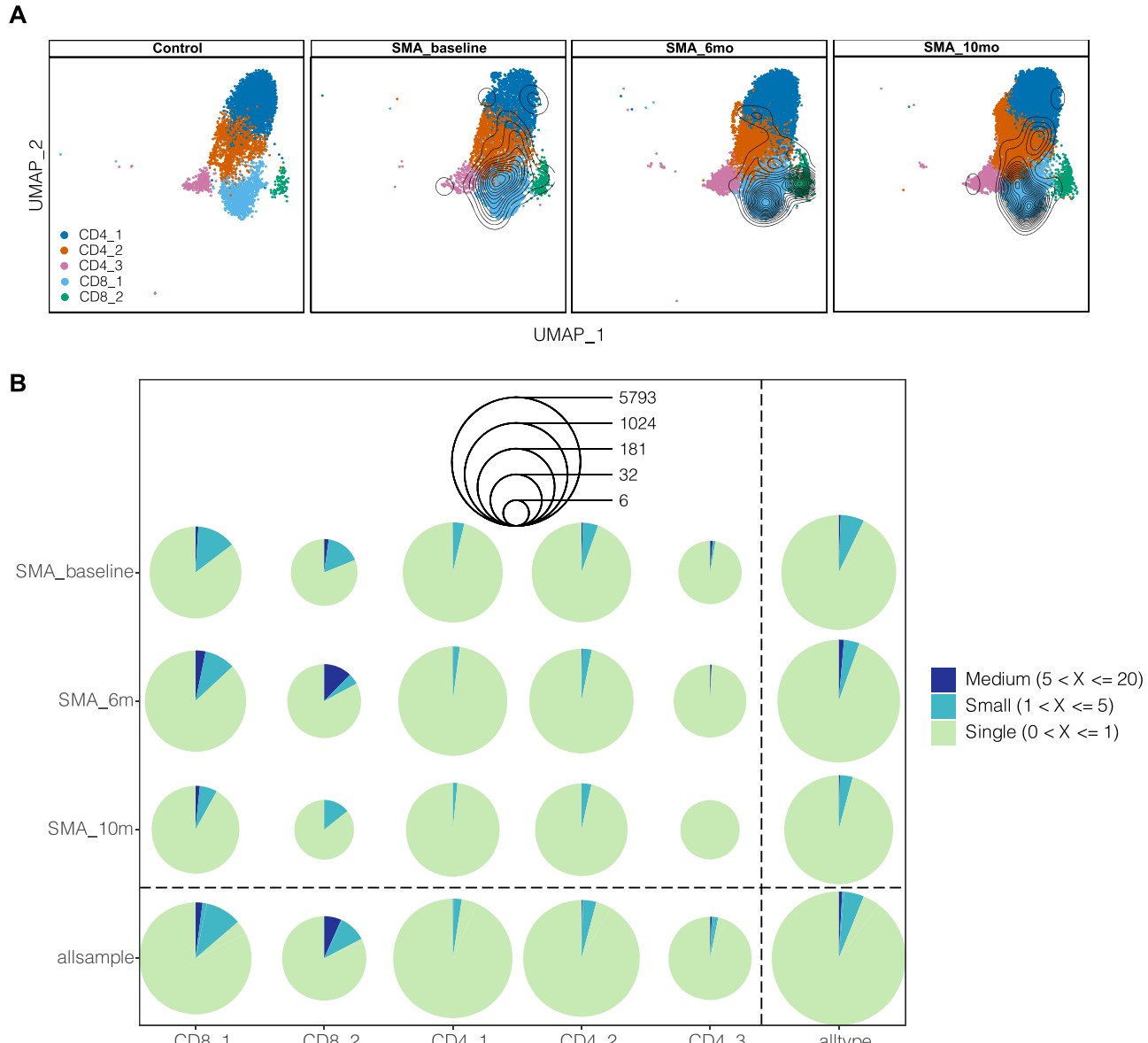

**Fig. 4 | Effect of nusinersen therapy in CSF T cell repertoire. A** UMAPs embedding of T cell subclusters (CD8_1, CD8_2, CD4_1, CD4_2 and CD4_3) from Control, SMA_baseline, SMA_6mo and SMA_10mo overlaid with contour plots corresponding to the projected density of T cells with more than five TCR clonotype count.

**B** Pie plots of clonal proportions based on TCR clonotype counts (Single: $0 < X \leq 1$; Small: $1 < X \leq 5$; Medium: $5 < X \leq 20$) within T cell subclusters at different time points following nusinersen treatment. The radiuses of the circles indicate "cell counts". Source data are provided as a Source Data file.

revealed a decreasing trend in cytotoxicity-associated proteins, including IL-18R1, CD5, PD-L1 and TNFRSF9, along with two other inflammatory proteins, CX3CL1 and MMP-1, after 6 months of treatment. Notably, these molecules were not part of Nuzzo et al.'s 27-plexed panel. Nevertheless, our findings align with insights derived from our CSF transcriptome data, collectively highlighting the neuroimmunomodulatory impact of nusinersen therapy.

IL-18 has been previously identified as a significant contributor to the development of neuronal injury[52–54]. Upon experimental axonal crush injury, IL-18 expression was significantly increased in injured nerves in both CNS and periphery, and infiltrating macrophages were identified as the primary cellular sources of elevated IL-18 levels within the first few days following axonal injury[53]. Furthermore, local resident microglia exhibited enhanced IL-18 expression mainly at sites of myelin

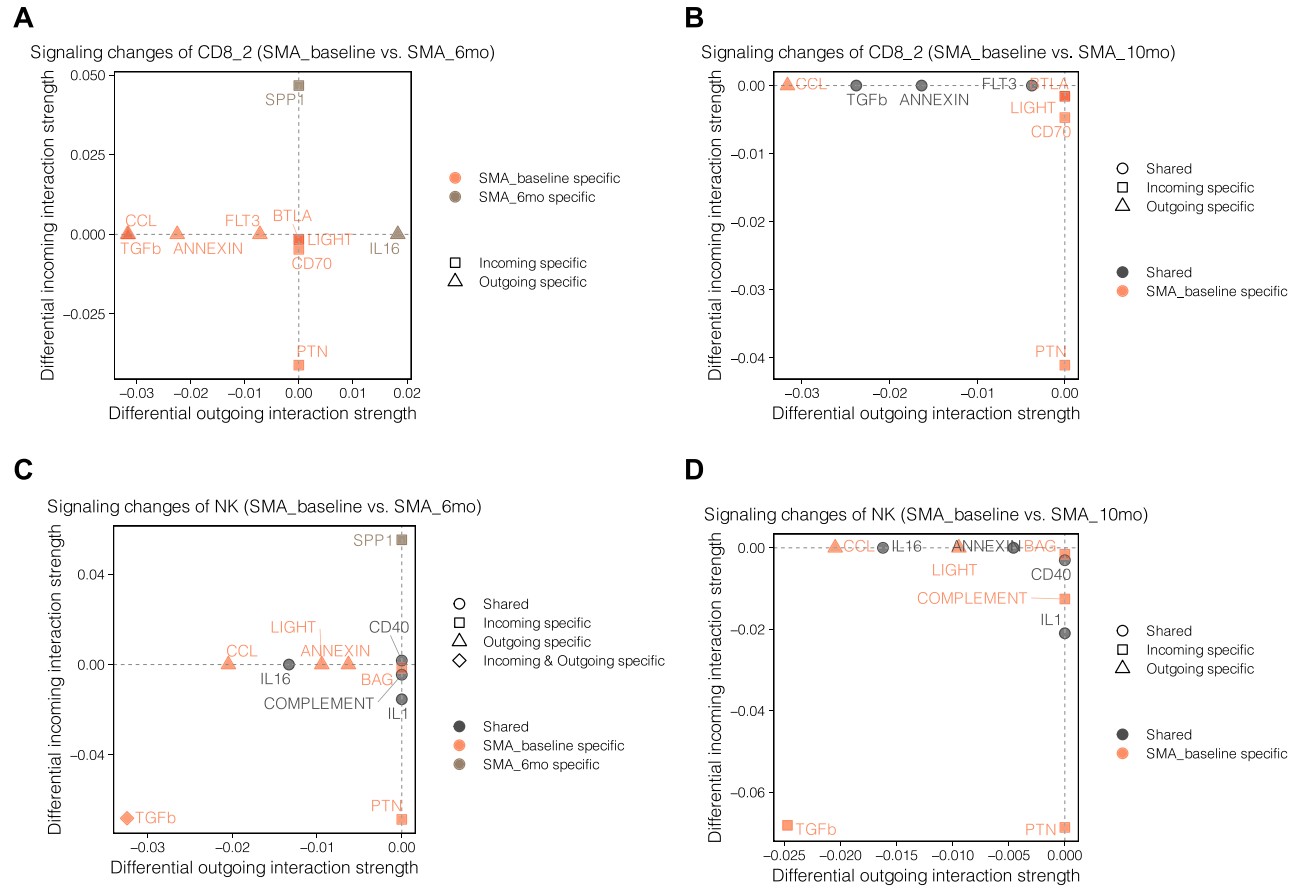

**Fig. 5 | Effect of nusinersen therapy in CSF cytotoxic cells.** Altered CD8_2-associated signaling pathways in SMA_baseline versus SMA_6mo (**A**) and SMA_baseline versus SMA_10mo (**B**). Altered NK-associated signaling pathways in SMA_baseline versus SMA_6mo (**C**) and SMA_baseline versus SMA_10mo (**D**).

degradation, suggesting the involvement of IL-18-mediated microglial neurotoxicity[53]. Collectively, these findings suggest the involvement of IL-18 in the cytokine network associated with the robust neuroinflammatory response that occurs after neuronal injury. Notably, prior investigations have shown an elevation in IL-18 upregulation in the CSF of sALS patients[55].

However, it remains possible that there are cells other than IL-18-secreting myeloid cells that functionally contribute to the induction of cytotoxic activities in SMA CNS tissues, even though other cell types expressed lower levels of IL-18. It is likely that other cells contribute to the regulation of cytotoxic immune responses through the secretion of other factors. Our current data are also insufficient to definitively ascertain whether immune cells, such as IL-18+ CD68+ myeloid cells, primarily reside in the CSF, originate solely within the CNS, migrate from the CNS to peripheral tissues, or emerge from the periphery with the intent of entering the CNS. To address these complexities, future research may benefit from employing murine SMA models. Such models would provide a robust platform to gain deeper insights into the origin and trafficking dynamics of cytotoxic CD8 T cells and IL-18-secreting monocytes within the context of SMA. Furthermore, our study is limited by the sample size and the observation time. Despite these limitations, our study provides new insights into the generalizable immune mechanisms involved in SMA and suggests that targeting these mechanisms may be a potential therapeutic approach for the disease.

## Methods

### Patient characteristics and ethics statements

All patients were recruited and all samples were collected at the Department of Neurology, University Hospital Essen and the Department of Neurology with Institute of Translational Neurology, University Hospital Münster. The study was approved by the local ethics committee in Essen (Ethics Committee of the University Duisburg-Essen; reference number 18-8285-BO) and in Münster (Ethics Committee of the Board of Physicians of the Region Westfalen-Lippe and of the Westfälische Wilhelms-University Münster; reference 2015-522-f-S). All patients gave written informed consent to sample collection and data analysis. Basic patient characteristics are provided in Table 1.

### Preparation of CSF cells and cell-free supernatant

Nusinersen was administered to SMA patients by lumbar puncture on days 1, 14, 28 and 63 followed by a repetitive application every 4 months according to the treatment guideline. Treatment was performed between the years 2017 and 2021 in the center in Essen. Before each nusinersen injection, the same amount (5 mL) of CSF was withdrawn to prevent excessive intra-CSF pressure. From these samples collected during clinical routine, CSF was processed for scientific purposes from 19 SMA patients ($N = 9$ in Cohort 1; $N = 10$ in Cohort 2; Table 1; Supplementary Data 1) at baseline before administering the first treatment and after 2 months ($N = 2$), 6 months ($N = 6$) and 10 months ($N = 8$) of intrathecal nusinersen treatment. Samples were discarded if blood contamination was visually detectable in the CSF sample. Nineteen patients diagnosed with IIH were chosen as age-matched controls ($N = 9$ in Cohort 1, whose data were published previously[33,34]; $N = 10$ in Cohort 2; Table 1; Supplementary Data 1). CSF cells were processed within 2 h post lumbar puncture to ensure optimal sample quality. For processing, CSF was collected into round bottom polypropylene tubes and then centrifuged for 10 min at $300 \times g$. The cell-free supernatants were carefully withdrawn with a

pipette, frozen and stored at −80 °C for subsequent proteomics. For scRNA-seq, the remaining CSF cell pellets were resuspended in 5 ml of X-Vivo15 media (Lonza) and stored on crushed ice. The samples were then transported with same-day shipping on crushed ice to the laboratory of the Department of Neurology with the Institute of Translational Neurology at the University Hospital Münster for scRNA-seq. Cell suspensions were then centrifuged again for 5 min at 400 × *g* and resuspended in 50 μl of X-Vivo15 media (Lonza). Out of this volume, 6 μl were used for cell counting and the remaining volume was entirely used for scRNA-seq.

## Single-cell sequencing and data preprocessing

Single-cell suspensions were loaded onto the Chromium Controller (10× Genomics) using the Chromium Single Cell 3′ GEM, Library & Gel Bead Kits v3 and v3.1 (10× Genomics). For TCR sequencing, we applied a previously described technique to sequence antigen receptor information from the 3′ scRNA-seq libraries[56] (Fig. S7A). In short, the method involves self-circulating the cDNA library, enriching the VDJ region and re-linearizing to remove the constant region of antigen receptors during enrichment, while maintaining their cell barcode and unique molecular identifier (UMI) information attached to the 3′ of the cDNA molecules. Library preparation was performed according to manufacturer's instructions using AMPure beads (Beckman Coulter). Sequencing was carried out on a local Illumina Nextseq 500/2000 and Novaseq6000 with a 28-8-0-57 or 28-8-0-91 read setup. Processing of these sequencing data together with the previously collected data from 9 control donors was performed with the CellRanger pipeline v6.1.0 (10× Genomics) according to the manufacturer's instructions. Briefly, raw BCL files were de-multiplexed using the CellRanger *mkfastq* pipeline. Subsequent read alignments and transcript counting was done individually for each sample using the CellRanger *count* and CellRanger *vdj* pipelines with standard parameters. Details regarding sequencing depth and cell recovery are provided in Supplementary Data 1.

## Single-cell analysis

We used Seurat v4.3[57] for the subsequent single-cell analysis. The cellranger matrix was loaded into Seurat. SoupX[58] 1.6.2 was carried out with the automatic method using clustering information to reduce ambient RNA expression. Doublets were identified with scDblFinder[59] 1.12 and default parameters. We removed doublets, genes with high mitochondrial percentages (5–15), few genes (<200) or high genes (1300–7500) for each sample manually based on manual inspection of the QC plots. Next, we normalized the samples separately with SCTransform v2[60] and then removed the batch effect with Harmony[61]. Clusters and UMAP embeddings were computed based on the Harmony reduction within Seurat using a resolution of 0.3 and 40 dimensions. The top markers of each cluster were determined with the RunPresto function, which uses a Wilcoxon test with Bonferroni correction, based on the log-normalized data. Clusters were annotated based on known literature genes and supported by automatic annotations of the PBMC reference in Azimuth[62]. Dot plots and feature plots were generated with Seurat functions based on the log-normalized data.

## Differential abundance (DA), differentially expressed genes (DEG) and GO enrichment analyses

Cluster abundance was determined with the propeller[63] tool, which is part of speckle v0.99. Briefly, we transformed the cluster proportions with a logit function. When comparing SMA to Control, we build a linear model adjusting for sex and age. When comparing different time points within SMA patients, we included the patient as a random effect in the linear model because of paired data with incomplete pairs. We computed significance based on the robust empirical Bayes method

from limma. *P* values were adjusted with the Benjamini–Hochberg method. The results were visualized with ggplot2.

DEG was determined with a pseudobulk approach from limma v3.54[64]. Briefly, we created a linear model as explained above in the cluster abundance analysis. Next, we applied the voomWithQualityWeights[65] function in limma. Significance was calculated again based on the robust empirical Bayes method from limma. Volcano plots were created using the EnhancedVolcano package. DEG (adjusted *p* value < 0.05, log2 fold change > 1) were tested for enrichment in pathways with enrichR[66] using the integrated library "GO Biological Process 2021" separately for up and downregulated genes.

## Single-cell immune repertoire analysis

We analyzed single-cell T cell receptor sequencing data with scRepertoire v1.7[67]. The filtered outputs of CellRanger vdj were imported, and the TCR heavy and light chains were combined in each cell based on their barcodes. Cells with more than two immune receptor chains were removed from the dataset. Clonotypes were called using the CDR3 amino acid sequence. Clonal frequency, shown in Supplementary Data 9, was overlaid with the UMAP embeddings with the clonalOverlay function in scRepertoire.

## Cell–cell communication analysis

CellChat[39] R package was applied to investigate and visualize the cell–cell communication and ligand–receptor interactions between different CSF cells. We followed the developers' instructions, briefly applied the standard package functions and used a subset of CellChatDB, namely "Secreted Signaling", for cell–cell communication analysis with default parameters to process the annotated and normalized scRNA-seq data. Interactions with fewer than 10 cells were filtered out, and the gene expression data were projected onto protein–protein interaction. netAnalysis_contribution function was applied to identify the significant ligand–receptor pair to a given signaling pathway. Details of the inferred cell–cell communications of Control, SMA_baseline and SMA_10mo are provided in Supplementary Data 4.

## Spatial gene expression assay and data preprocessing

Four post-mortem formalin-fixed paraffin-embedded (FFPE) brain samples with acute infantile SMA type I were obtained in University Hospital Münster (*N* = 1) and Medical University of Vienna (*N* = 3; Supplementary Data 10), approved by the Ethics Committee of the Medical University of Vienna (EK 1454/18; 1636/2019). Immunohistochemical staining for CD8 (clone C8/144B, 1:100, Agilent, GA62361-2) and Caspase-3 (clone 5A1E, 1:100, Cell Signaling Technology, 9664) was performed using the streptavidin–biotin method on an automated staining system (DAKO OMNIS, Agilent). For the spatial gene expression assay, the 5-μm FFPE sections were placed on the fiducial frame of the capture area on the Visium Spatial Gene Expression slides and processed according to the manufacturer's instructions (10× Genomics). Brightfield histological images following H&E staining were taken using a 10× objective on the Nikon Eclipse Ni scanner. Next-generation sequencing libraries with dual indexes were prepared according to the Spatial Gene Expression Slide Kit user guide (10× Genomics), and sequenced on a local Illumina Nextseq 2000 with a 28-10-10-90 read setup. Raw BCL files were de-multiplexed using the Spaceranger *mkfastq* pipeline v2.0.0 (10× Genomics). Subsequent fiducial alignment and tissue detection on the brightfield image input as well as read alignment to the Visium_Human_Transcriptome_Probe_Set_v1.0_ GRCh38-2020-A and transcript counting were done using the SpaceRanger *count* pipeline with standard parameters. Feature plots were generated with Seurat SpatialFeaturePlot function.

## Multiplexed immunofluorescence (IF) histological staining of SMA brain tissue

Multiplexed immunohistochemistry was performed on the above-mentioned FFPE samples of four post-mortem brain tissues of infants with SMA type I and two sex- and age-matched control brain tissues. Both control brain tissues had no neuropathy reported. Multiplexed IF was performed using the Opal multiplex system (Akoya Biosciences) according to the manufacturer's instruction[68]. In brief, FFPE sections of 2-μm thickness were deparaffinized and then fixed with 4% paraformaldehyde prior to antigen retrieval by heat-induced epitope retrieval using citrate buffer (pH6) or Tris/EDTA (pH9). After cooling down from antigen retrieval, slides were immersed in alkaline hydrogen peroxide solution (pH11.5) in transparent plasticware and exposed to white light by sandwiching the immersed slides between two light-emitting diode (LED) panels. The slides were incubated for 45 min, after that the solution was discarded and replaced by freshly prepared hydrogen peroxide solution for another 45 min LED exposure. Slides were then washed with PBS and ready for subsequent staining. Each section was put through several sequential rounds of staining (Supplementary Data 11); each included endogenous peroxidase blocking and non-specific protein blocking, followed by primary Ab and corresponding secondary horseradish peroxidase-conjugated polymer (Zytomed Systems, Germany or Akoya Biosciences). Each horseradish peroxidase-conjugated polymer mediated the covalent binding of different fluorophores using tyramide signal amplification. Such covalent reaction was followed by additional antigen retrieval in heated citric buffer (pH6) or Tris/EDTA (pH9) for 10 min to remove antibodies before the next round of staining. After all sequential staining reactions, sections were counterstained with DAPI (Vector Lab). The sequential multiplexed staining protocol is shown in Supplementary Data 11. Slides were scanned and digitalized using a Zeiss AxioScanner Z.1 (Carl Zeiss AG, Germany) with 10× objective magnification. Quantification of co-expressing markers and proximity analysis were performed with HALO (Indica Labs).

## CSF protein quantification and functional enrichment analysis

For unbiased proteome analysis, 10 μL of cell-free CSF supernatant from the donors in Cohort 2 (SMA_baseline: $N = 10$; SMA_6mo: $N = 10$; Control: $N = 10$; Table 1) were diluted with 190 μL lysis buffer (8 M urea, 2% sodium dodecyl sulfate, 100 mM TrisBase, 10 mM tris (2-carboxyethyl) phosphine) and prepared for HDMS analysis by filter-based reduction, alkylation and tryptic digestion[69]. Peptides were dissolved in 20 μl 0.1% formic acid containing 5% acetonitrile and 3 μl was analyzed by reversed-phase liquid chromatography coupled to HDMS with Synapt G2 Si/M-Class nanoUPLC (Waters Corp., Manchester, UK) using C18 μPAC columns (trapping and 50 cm analytical; PharmaFluidics, Ghent, Belgium) with a 90 min gradient (solvent system 100% water versus 100% acetonitrile, both containing 0.1% formic acid). Samples that showed low resolution and signs of residual polymer in liquid chromatography did not pass the quality control and were not used for the comparative data analysis. Consequently, data of 7 Control, 8 SMA_baseline and 5 SMA_6mo samples were analyzed. Differentially abundant proteins are provided in Supplementary Data 4. Statistical Analysis was performed using R 3.5.4 and Progenesis QIP (Waters Corp.). Differences between SMA_baseline and SMA_6mo were analyzed using paired $t$-test; differences between Control and SMA_baseline/SMA_6mo were analyzed using ANOVA test[70]. Enrichment of DE proteins was performed as described above for genes using enrichR and the internal library "GO_Biological_Process_2021". The mass spectrometry proteomics data have been deposited to the ProteomeXchange Consortium via the PRIDE[71] partner repository with the dataset identifier PXD051600.

For targeted CSF proteome analysis, additional 92 predefined proteins were quantified in aliquots of the same 30 CSF samples using the Olink Target 96 Inflammation kit according to manufacturer's instructions (Olink Target 96 Inflammation, Olink). It is based on the PEA technology that obtains protein information by linking protein-specific antibodies to DNA-encoded tags that were subsequently detected and quantified using standard real-time quantitative polymerase chain reaction. Statistical analyses were performed by using Olink® Insights Stat Analysis APP, and protein abundance values are shown in the Normalized Protein eXpression unit, an Olink's arbitrary unit in log2 scale (Supplementary Data 8).

## Statistics & reproducibility

The sample size in this study was not predetermined using statistical methods. Availability of patient-derived samples constrained our sample size, which was deemed adequate based on comparisons with similar studies in the field. Three CSF samples from SMA patients were excluded from scRNA-Seq due to visible blood contamination. All acquired scRNA-Seq data were included in the analyses. This study was exploratory and did not involve randomization of patients or samples. Samples were processed without blinding but were pseudonymized, minimizing the potential for bias.

## Reporting summary

Further information on research design is available in the Nature Portfolio Reporting Summary linked to this article.

## Data availability

Source data are provided with this paper, and further data are available in Supplementary Data 1–12. All single-cell sequencing data including metadata regarding cluster and treatment are available in the Gene Expression Omnibus (GEO) repository under reference number GSE232391. The mass spectrometry proteomics data are available via ProteomeXchange with identifier PXD051600.

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

## Acknowledgements

We express our sincere gratitude to Julia Tietz, Rebecca Ley, Doreen Ackermann, Anna Maria Berg, Luzia Buchholz and Irene Erber for their exceptional technical support throughout this project. We also extend our appreciation to Linda-Isabell Schmitt, Stefanie Hezel and Christina David for their efforts in preparing PBMCs. Furthermore, we are deeply thankful to the CSF Laboratories at University Hospital Essen and University Hospital Münster for their outstanding management of the patient specimens. Finally, we extend our heartfelt thanks to all the patients who generously participated in this study. This study was in part funded by Biogen Inc. in the context of a sponsored research agreement. The sponsor had no involvement in the design of the study or in the analysis or presentation of the data. M.H. and G.M.z.H. were supported by the Interdisciplinary Center for Clinical Research (IZKF) of the medical faculty of Münster (MzH3/020/20 to G.M.z.H. and SEED/016/21 to M.H.). G.M.z.H. was supported by grants from the Deutsche Forschungsgemeinschaft (DFG) (ME4050/12-1, ME4050/13-1) and by a grant from the Bundesministerium für Bildung und Forschung (BMBF) "*Lipid Immune Neuropathy Consortium*".

## Author contributions

G.M.z.H. and T.H. coordinated and supervised all aspects of the study. I.N.L., P.F.Y.C. and M.H. conducted and analyzed the majority of experiments. C.T., G.G., C.H., J.T.S. and R.H. contributed FFPE samples and participated in the acquisition and interpretation of histology data. T.W., C.A.D. and L.K. facilitated equipment provision and conducted flow cytometry analysis. S.K., C.B.S., C.J.N. and T.R. carried out HDMS data acquisition and analysis. M.L., M.E., C.K., H.W. and T.H. provided CSF samples and clinical data. I.N.L., P.F.Y.C., M.H. and G.M.z.H. collaborated on writing the manuscript, with input and significant revisions from all authors.

## Funding

## Competing interests

G.M.z.H. received compensation for serving on scientific advisory boards (LFB, Roche, Immunovant) and speaker honoraria (Alexion, LFB, Argenx). H.W. is acting as a paid consultant for AbbVie, Actelion, Biogen, IGES, Johnson & Johnson, Novartis, Roche, Sanofi-Aventis and the Swiss Multiple Sclerosis Society. C.K. received lecture and consultancy fees from Biogen, Roche and Novartis. T.H. received lecture and consultancy fees from Biogen, Roche and Novartis, as well as research support from Biogen, Roche and Novartis Gene Therapies. The remaining authors declare no competing interests.
