## [Peer Review File · Nature Communications]

Cell-mediated Cytotoxicity within CSF and Brain Parenchyma
in Spinal Muscular Atrophy Unaltered by Nusinersen
TreatmentReviewers' comments:

Reviewer #1 (Remarks to the Author):

Spinal muscular atrophy (SMA) is a motoneuron disease caused in 94% of cases by a loss of function of the SMN1 gene. In this manuscript, Lu et al set out to analyze the immune repertoire in the CSF of SMA patients before and after treatment with nusinersen, an antisense oligonucleotide (ASO) which is administered intrathecally into the CSF to increase compensatory expression of the orthologous SMN2 gene. Using single-cell RNA and TCR-seq, the authors found that although nusinersen did not affect the immune cell phenotype and composition in SMA patients, they identified an expansion of cytotoxic NK cells and CD8+ T cells in baseline conditions in SMA patients compared to controls. This was confirmed using spatial transcriptomic profiling coupled with multiplex immunohistochemistry in an SMA-affected brain. Overall, the manuscript provides valuable insight into the inflammatory profile of the SMA CSF. Although the authors begin to link these changes to motoneuron degeneration in SMA patients, the sample size is small, as low as an 'n' of 1.

Histological analyses do make up for these shortcomings by validating ligand/receptor interactions found in the CSF. While the CSF proteomics data provide a valuable platform for biomarker discovery, the data do not support the differential CellChat findings discovered over the course of treatment. The authors make a valid point that this could be due to the CSF being collected very early in the duration of treatment and therefore may not have had enough time to change meaningfully. However, it draws into question the practical validity of the observed transcriptional changes. Thus, protein-based validation using flow cytometry or other related measures is necessary to validate these findings.

Figure 3A: Line 228 states that granzyme genes are located in close proximity to DeN. While this may be true, the number of spots corresponding to GZMB is very small and could be the result of sequencing noise. To this end, the highest expressing spot for GZMB is in the same location as the highest expressing spot for C1QL1 and IL18. Additionally, H&E stained slides of the same section used for spatial transcriptomics would be helpful in understanding the tissue architectural context, especially if the inflammatory signature is oriented close to blood vessels and other features.

Line 248: It is unclear from the methods section how the TCR information was retrieved from 3' sequencing data. Although the methods reference a paper from 2021 in PNAS that reportedly performed a similar analysis, the method is also not described in the methods section of this reference.

Figure 4A: Highlighting the cells and annotating by expansion threshold rather than using density plots would be more clear and easier to interpret. As is, it is difficult to interpret which cells/clusters are expanded, especially in the CD4 populations.

Line 251: What measure was used to determine the effects on T cell expansion? Was it the level of expansion of the top clone or a global measure of diversity (i.e. Shannon entropy)? Was this performed on a per-cluster basis? It looks like at 10mo there may be a reduction in expansion of CD8_2, although it

also increased at 6mo in comparison to baseline.

CellChat was performed on cells in the CSF including monocytes and T cells. Cells resembling border-associated macrophages (BAM) were also found in the CSF. Are these derived from the puncture itself or are there BAMs/pre-BAMs circulating through the CSF? Likewise, ligand/receptor pairs including IL18-mediated signaling from CSF monocytes are speculated to recruit T cells to the CNS. Meanwhile, staining in Figure 3 highlights IL18 secreted by CD68+ cells. Are these CD68+ cells monocyte-derived and engrafted from the CSF or are they microglia? If they are microglia, it is unclear how the CSF and brain data are related.

Reviewer #2 (Remarks to the Author):

Nature Communications NCOMMS-23-26631

“Cell-mediated cytotoxicity surrounding the brain in spinal muscular atrophy”

These authors present a well-written and sufficiently detailed report on immune responses in CSF from patients with SMA treated with nusinersen. The results are presented clearly in several figures. Single cell RNA-sequence transcriptomic and mass spectroscopy proteomic studies demonstrated an expansion of cytotoxic NK cells and CD8+ T cells in nine untreated adult SMA patients (mainly type III and 3 or 4 copies of SMN2), possibly activated by IL-18- and CCL5- secreting monocytes, yet without discernable alterations in these same patients once treated with nusinersen. CSF from 19 patients with idiopathic intracranial hypertension were used as a control. Interestingly, multiplex immunohistochemistry studies in a post-mortem spinal cord specimen from a single nusinersen-treated patient (presented as a type I, 3 days old) demonstrated cytotoxicity in the vicinity of degenerating/dying motoneurons, differing from two controls. The authors postulate that the pathobiology of SMA includes an intrinsic cytotoxic response that contributes to motor neuron pathology, and that nusinersen treatment did not mitigate this.

Comments:

1. Title: the word “surrounding” is confusing. Consider changing to a more relevant descriptor of your work, such as “within CSF and brain parenchyma” or similar.
2. Introduction: while this report focuses on the immune response in SMA, the authors may wish to add a brief summary of published human post-mortem studies that describe possible inflammation related changes (gliosis) in the CNS in SMA patients. This would supplement the animal data cited. Reference 12 focuses upon splenic hypoplasia in the Taiwanese mouse model of SMA and in 9 post-mortem spleen samples from type 1 patients. For additional consideration regarding neuropathological findings of gliosis: (1) Araki, S et al. Neuropathological analysis in spinal muscular atrophy type II. *Acta Neuropathol* 106, 441–448 (2003), and Kuru, S et al. (2009), An autopsy case of spinal muscular atrophy type III (Kugelberg-Welander disease, *Neuropathology*, 29: 63-67), and several older neuropathology case series. It would also be useful to mention the current consideration that SMA is a non-cell autonomous disease

where inflammatory changes appear to precede motor neuron loss (e.g., Abati et al, Glial cells involvement in spinal muscular atrophy: Could SMA be a neuroinflammatory disease? 2020, *Neurobiology of Disease*, 140).

3. Introduction: it is necessary to discuss additional relevant papers in the literature, at a minimum: (1) Nuzzo et al 2023 (Nusinersen mitigates neuroinflammation in severe spinal muscular atrophy patients, PMID: 36792810) , (2) Bonanno et al 2022 (Identification of a cytokine profile in serum and cerebrospinal fluid of pediatric and adult spinal muscular atrophy patients and its modulation upon nusinersen treatment, PMID 36035258) and (3) Freigang et al. (Increased chitotriosidase 1 concentration following nusinersen treatment in spinal muscular atrophy, PMID: 34321067). Nuzzo focuses on CSF pro- and anti-inflammatory cytokines in type 1 patients, and compared to types 2 and 3, and the response to nusinersen therapy. Notably, they do not identify IL-18 as an abnormally elevated cytokine in the severe type 1 patients.

4. Introduction: consider mentioning that nusinersen does not apparently generate anti-drug antibodies, at least no evidence for neutralizing antibodies in CSF or blood has been reported.

5. Methods: suggest adding to table 1 the results of the routine CSF analyses – red and white cell counts, protein level. This will aide in demonstrating the presence or absence of an obvious inflammatory situation, as has been reported previously in some nusinersen-treated patients (apparently without clinical symptomatology). Reference 18 is cited but is an over-simplification of the CSF findings in nusinersen treated patients. They report increased CSF leukocytes in 3 of 72 patients with SMA type 1 treated with nusinersen – infrequent but not rare (despite those authors’ characterization).

6. Results: pro- and anti-apoptotic drivers are mentioned by the authors. Were more specific studies of neuronal apoptosis performed? This has been addressed to a limited degree in earlier literature.

7. Discussion: it would be helpful to address the limitations of linking CSF data from adult SMA patients with neuroimmunopathology in the spinal cord of a single very severe SMA neonate (probably a type 0?). As in point 3 above, your observations and interpretation need to be discussed in some detail in relation to the Nuzzo findings – where are you in alignment with those earlier findings and where are you presenting new and different or contradictory findings, and how these may relate to the SMA type and the age and severity of disease. Do you believe the observations in CSF in adult and less severely affected SMA patients are likely similar to that expected in more severely affected infants? Is your data congruent with other studies in the literature?

Reviewer #3 (Remarks to the Author):

The manuscript describes an analysis of CSF fluid and tissue from adult SMA patients using single cell, single nucleus, and spatial transcriptomics. The authors find that SMA patients when compared to healthy controls have an enrichment of activated and cytotoxic T cells in their CSF at baseline, prior to ASO treatment. ASO treatment did not alter the observed inflammatory profile, which might suggest that this profile is more cause than consequence of SMA. The inflammatory profile appeared to have an enrichment of IL-18 signaling pathways. TCR analysis showed increased clonal expansions in SMA patients. Spatial analysis found consistent results with the transcriptional profile, and observed activated

CD8 T cells in proximity to neurons. This analysis was in a single SMA patient, with five healthy controls. The manuscript is clearly written and identifies a detailed profile of cell-mediated inflammation that may be contributing to SMA etiology. The N is low, but this is sufficiently acknowledged throughout and consistent with analyses of rare diseases.

1) The authors focus quite a bit on CD8_2, but they don't describe what CD8_1 is (and from the limited info given, it seems like it might also be cytotoxic). In the later clonal analysis CD8_1 I think has the largest clonotypes which would also be a point of interest. Clearer DEG analysis between CD8_1 and 2 would be helpful (especially because at least at a glance CD8_1 does not appear naïve).

2) The authors make claims about the abundance of cells between SMA and healthy controls (e.g. line 182), but it's not clear they've provided the right info for this analysis. Rather than just comparing frequencies with the 10X object, they need to compare some total number or absolute frequency of cells back calculated based on the amount of cells that were put into the 10X pipeline and successfully recovered.

3) A related problem occurs on lines 251—"T cell expansion" is not really what's being measured here, but rather the number of observed clonal expansions. This could have sampling bias as above and should be carefully controlled for based on cells input and recovered. It would be great to have clonal expansions plotted as a chart across samples.

4) The TCR analysis is underutilized. First, figure 4A is somewhat difficult to interpret. Why not use a standard measure of clonality (that accounts for sampling) such as a diversity index like Shannon or Simpson's?

5) A second simple TCR analysis would be to compare clonal lineages between time points—are the same cells present in individuals during treatment? Are the expanded clones consistently observed? This would be important info to highlight. This information is available in Supp Table 9 but as it is a sparse table it's difficult to scan to get an impression of the data structure.

6) Last, running some additional (simple) TCR analyses would be informative—first, looking for clusters of related TCR sequences can easily be done by free tools (tcrdist, GLIPH) or even just applying a simple edit distance. Additionally, trying to match the observed sequences to public databases might be informative if a match is found. This can be difficult given HLA restrictions and the limited nature of public databases (e.g. vdjdb) but as the authors note, a prior study in Alzheimers found expanded CD8 T cells that were able to be mapped to an EBV specific response.

Cell-mediated Cytotoxicity within CSF and Brain Parenchyma in Spinal Muscular Atrophy

POINT BY POINT RESPONSE TO REVIEWERS

Reviewer #1

General comments

Overall, the manuscript provides valuable insight into the inflammatory profile of the SMA CSF. Although the authors begin to link these changes to motoneuron degeneration in SMA patients, the sample size is small, as low as an 'n' of 1.

Response: We acknowledge the initial limitation in our sample size, which stemmed from challenges in accessing suitable SMA-derived brain biomaterial. Despite reaching out to multiple international biobanks, our efforts were initially unsuccessful.

However, we are pleased to inform the reviewer that, subsequent to this feedback, we undertook further endeavors. We have successfully obtained three additional brain tissues from infants with SMA type I, courtesy of Prof. Romana Höftberger at the Medical University of Vienna who is now included as co-author. This collaboration has expanded our sample size to a total of four available SMA brain specimens.

This increased sample size has allowed us to delve deeper into our investigation, confirming the presence of cytotoxic CD8 T cells in SMA-affected brains through immunohistochemistry (IHC) and multiplex immunofluorescence (Supplementary Figure 5). Notably, we have also validated the spatial proximity of these cytotoxic CD8 T cells to IL-18-secreting CD68+ myeloid cells, as outlined in the updated data presented in Figure 3F-G and the corresponding results section on page 8 of the manuscript. In fact, CD8+ cells were promptly identified 'at first glance' in all brainstem material from type 1 SMA patients. We are thus confident that our observations of intraparenchymal cytotoxicity represent a common feature of SMA patients and are not just a single-patient 'outlier observation'.

Supplementary Figure 5:

Figure 3F, 3G:

Histological analyses do make up for these shortcomings by validating ligand/receptor interactions found in the CSF. While the CSF proteomics data provide a valuable platform for biomarker discovery, the data do not support the differential CellChat findings discovered over the course of treatment. The authors make a valid point that this could be due to the CSF being collected very early in the duration of treatment and therefore may not have had enough time to change meaningfully. However, it draws into question the practical validity of the observed transcriptional changes. Thus, protein-based validation using flow cytometry or other related measures is necessary to validate these findings.

Response: We appreciate the reviewer's thoughtful feedback. During our analysis of the CSF proteomics data, we indeed observed a consistent trend indicating a reduction in cytotoxicity-associated proteins such as IL-18R1, CD5, PD-L1, and TNFRSF9 after 6 months of Nusinersen treatment. This observation aligns with our transcriptome findings. Furthermore, we noted reductions in two other inflammatory proteins, namely CX3CL1 and MMP-1, following the treatment.

It is also important to highlight that while these reductions in protein levels demonstrated statistical significance when assessed using the Wilcoxon signed-rank test, we also have considered correction for multiple hypothesis testing¹, given that we measured a total of 92 proteins through the Olink PEA technique. To address this, we applied the Benjamini-Hochberg method to control the false discovery rate, as well as the family-wise error rate methods of Holm's and Hochberg's for the adjustments to the p-values. After applying these multiple hypothesis-testing corrections, the statistical significance of the treatment-associated reductions in cytotoxic and inflammatory proteins did not persist (Table_S8). Despite this formal non-significance, the overall trend toward reduced cytotoxic and inflammatory CSF proteins following treatment remains consistent and we have discussed this on page 9 of the manuscript.

We acknowledge that we should have provided a more comprehensive explanation of our CSF proteomics data, instead of simply stating that we did not observe any changes. In light of this valuable feedback, we have expanded our observations in the Results section and Methods section and have included Supplementary Figure 9 (reproduced below) to provide further clarification. We hope these additions will address the reviewer's concerns.

Supplementary Figure 9:

Figure 3A: Line 228 states that granzyme genes are located in close proximity to DeN. While this may be true, the number of spots corresponding to GZMB is very small and could be the result of sequencing noise. To this end, the highest expressing spot for GZMB is in the same location as the highest expressing spot for C1QL1 and IL18. Additionally, H&E stained slides of the same

section used for spatial transcriptomics would be helpful in understanding the tissue architectural context, especially if the inflammatory signature is oriented close to blood vessels and other features.

Response: We have indeed acquired and thoroughly examined the H&E stained slide derived from the same tissue section utilized for spatial transcriptomics and we depict these images in Supplementary Figure 4F for reference. These additional data have provided a deeper understanding of the tissue's architectural context. Upon analysis of the H&E stained slide, we observed blood vessels in close proximity to DeN, which we indicate by the black arrows in Supplementary Figure 4F (reproduced below). This spatial arrangement suggests a potential mechanism wherein increased blood vessels in these regions may facilitate the access of cytotoxic cells, proteins, and other relevant substances to neurons.

Notably, while the number of spots corresponding to GZMB may appear small, we wish to emphasize that each Visium spot encompasses a diameter of 55 μm , thus accommodating multiple cytotoxic cells. Widely commercially available spatial transcriptomic techniques have not yet reached spatial resolution at single cell level.

We have taken the reviewer's advice and incorporated this finding into our manuscript. To provide a clearer and more comprehensive presentation of this crucial contextual data, we have introduced Supplementary Figure 4F to visually illustrate these observations.

Supplementary Figure 4F:

Line 248: It is unclear from the methods section how the TCR information was retrieved from 3' sequencing data. Although the methods reference a paper from 2021 in PNAS that reportedly performed a similar analysis, the method is also not described in the methods section of this reference.

Response: We previously developed and published a technique for retrieving full length mRNA molecule information from 3' scRNA-seq libraries including TCR information^{2,3}. This technique uses a circulation approach to shorten the part of the mRNA sequence that is near the cell barcode (e.g. constant region of the TCR). This brings the distant-from-barcode region of the mRNA within range of regular short range sequencers. In fact, this approach was recently commercialized by others as a pre-composed kit (<https://singleron.bio/product/detail-14.html>) albeit without our involvement. We reproduce the illustrative experimental scheme published previously in Supplementary Figure 7A. This outlines the process of extracting TCR information

from 3' sequencing data. We recognize the significance of methodological clarity and we appreciate this feedback as it aids us in achieving this objective.

Supplementary Figure 6A:

Figure 4A: Highlighting the cells and annotating by expansion threshold rather than using density plots would be more clear and easier to interpret. As is, it is difficult to interpret which cells/clusters are expanded, especially in the CD4 populations.

Response: We appreciate this valuable suggestion. To enhance the clarity and interpretability of our data, we have created pie plots that illustrate the expansion of cells within various T cell populations at different time points following Nusinersen treatment (as below, the radiiuses of the circles indicate 'cell counts'). These pie plots, now incorporated into Figure 4B, provide a more straightforward and informative representation of the expanded cell clusters, particularly within the CD4 populations. We refer to these plots on page 8 of the manuscript.

Figure 4B:

Line 251: What measure was used to determine the effects on T cell expansion? Was it the level of expansion of the top clone or a global measure of diversity (i.e. Shannon entropy)? Was this performed on a per-cluster basis? It looks like at 10mo there may be a reduction in expansion of CD8_2, although it also increased at 6mo in comparison to baseline.

Response: We appreciate the reviewer's astute observation concerning the fluctuations in CD8_2 T cell expansion at 10 months, as well as the increase at 6 months when compared to the baseline. In Figure 4B, we have presented the TCR clonal proportion and cell counts on a per-cluster basis. To offer a comprehensive view, we also conducted clonal proportion analysis, which assesses the proportion of specific clonal types in relation to their counts (Supplementary Figure 7E and 7F). Furthermore, we applied the true diversity index for estimating diversity, as illustrated in Supplementary Figure 7B. This choice of the true diversity index is deliberate, as it is designed to provide a more accurate representation of diversity in a dataset, like our TCR data. In contrast to traditional diversity indices, such as the Shannon or Simpson indices, which primarily emphasize richness (i.e., the number of distinct entities) and evenness (i.e., the distribution of entities within a population), we find that the true diversity index offers a more comprehensive measure that captures the entire spectrum of diversity. In summary, our TCR analyses did not reveal any significant differences between Control and SMA samples, nor did they indicate variations in SMA samples before and after Nusinersen treatment. These findings remain consistent across the diverse metrics we applied.

Supplementary Figure 7B:

Supplementary Figure 7E, 7F:

CellChat was performed on cells in the CSF including monocytes and T cells. Cells resembling border-associated macrophages (BAM) were also found in the CSF. Are these derived from the puncture itself or are there BAMs/pre-BAMs circulating through the CSF? Likewise, ligand/receptor pairs including IL18-mediated signaling from CSF monocytes are speculated to recruit T cells to the CNS. Meanwhile, staining in Figure 3 highlights IL18 secreted by CD68+ cells. Are these CD68+ cells monocyte-derived and engrafted from the CSF or are they microglia? If they are microglia, it is unclear how the CSF and brain data are related.

Response: We appreciate the insightful questions raised by the reviewer. Regarding the presence of BAMs/pre-BAMs in the CSF, it is notable that the presence of cells transcriptionally resembling border associated macrophages and microglia-like cells have been described repeatedly in scRNA-seq studies of the CSF^{3,4}. CSF BAMs are also observed across CSF studies^{5,6}. It is generally believed in the field that peripheral myeloid cells acquire a microglia- or border-like phenotype when reaching the CSF compartment. Strongest evidence supporting a peripheral origin stems from a human bone marrow transplant recipient, whose CSF cells were genetically confirmed to be graft-derived and not host-derived⁷. The CSF environment thus appears to imprint such a BAM/microglia-like phenotype on myeloid cells.

Our GO enrichment analysis indicates that these BAM-like clusters, namely CSF_M1 and CSF_M2, exhibit significant enrichment in biological functions related to neuroinflammatory responses, which suggests a potential connection to CNS function. Furthermore, their enrichment in GO functions related to leukocyte migration, epithelial cell migration, endothelial cell migration, and smooth muscle cell migration, underscoring their inherent trafficking and diapedesis capabilities. (see Fig.1A for Reviewers only below).

In addressing the nature of CD68+ myeloid cells in the brain, our single nuclei pathology sequencing (snPATHO-Seq) data reveals that a few of the *CD68* gene expressing cells are within the Microglia cluster (see Fig.1B for Reviewers only below). However, we acknowledge that monocytes can assume a microglial-like phenotype in the context of diseased brains, making it challenging to definitively ascertain their origin⁸. Thus, we cannot exclude the possibility that these CD68+ myeloid cells may be monocyte-derived or have migrated from the CSF. We discuss this aspect on page 11 of the manuscript.

A**B**
Figure 1 for Reviewers only: (A) GO enrichment terms for selected biological processes enriched in the DEG in CSF_M1 and CSF_M2 clusters. (B) Feature plots of *P2RY12*, *CD68*, *PTPRC*, *IL18*, and *CD14* genes and an UMAP representation of broad cell type classes (lower right) of the snPATHO-Seq dataset.

Reviewer #2

General comments

These authors present a well-written and sufficiently detailed report on immune responses in CSF from patients with SMA treated with nusinersen. The results are presented clearly in several figures.

Response: We would like to express our sincere gratitude for the reviewer's thoughtful and encouraging assessment of our manuscript.

Comments: 1. Title: the word "surrounding" is confusing. Consider changing to a more relevant descriptor of your work, such as "within CSF and brain parenchyma" or similar.

Response: We thank the reviewer for the thoughtful feedback. We propose the revised title: "Cell-mediated Cytotoxicity within CSF and Brain Parenchyma in Spinal Muscular Atrophy" for our manuscript.

2. Introduction: while this report focuses on the immune response in SMA, the authors may wish to add a brief summary of published human post-mortem studies that describe possible inflammation related changes (gliosis) in the CNS in SMA patients. This would supplement the animal data cited. Reference 12 focuses upon splenic hypoplasia in the Taiwanese mouse model of SMA and in 9 post-mortem spleen samples from type 1 patients. For additional consideration regarding neuropathological findings of gliosis: (1) Araki, S et al. Neuropathological analysis in spinal muscular atrophy type II. Acta Neuropathol 106, 441–448 (2003), and Kuru, S et al. (2009), An autopsy case of spinal muscular atrophy type III (Kugelberg-Welander disease, Neuropathology, 29: 63-67), and several older neuropathology case series. It would also be useful to mention the current consideration that SMA is a non-cell autonomous disease where inflammatory changes appear to precede motor neuron loss (e.g., Abati et al, 2020, Neurobiology of Disease, 140).

Response: We thank the reviewer for pointing this out. We have supplemented these important references in the text on page 4 and page 5 of the manuscript.

3. Introduction: it is necessary to discuss additional relevant papers in the literature, at a minimum: (1) Nuzzo et al 2023 (Nusinersen mitigates neuroinflammation in severe spinal muscular atrophy patients, PMID: 36792810) , (2) Bonanno et al 2022 (Identification of a cytokine profile in serum and cerebrospinal fluid of pediatric and adult spinal muscular atrophy patients and its modulation upon nusinersen treatment, PMID 36035258) and (3) Freigang et al. (Increased chitotriosidase 1 concentration following nusinersen treatment in spinal muscular atrophy, PMID: 34321067). Nuzzo focuses on CSF pro- and anti-inflammatory cytokines in type 1 patients, and compared to types 2 and 3, and the response to nusinersen therapy. Notably, they do not identify IL-18 as an abnormally elevated cytokine in the severe type 1 patients.

Response: We thank the reviewer for the important feedback and now cite and discuss these studies in the text on page 4 and page 5 of the manuscript.

4. Introduction: consider mentioning that nusinersen does not apparently generate anti-drug antibodies, at least no evidence for neutralizing antibodies in CSF or blood has been reported.

Response: We have considered the reviewer's feedback and mentioned it in the text on page 4 of the manuscript.

5. Methods: suggest adding to table 1 the results of the routine CSF analyses – red and white cell counts, protein level. This will aid in demonstrating the presence or absence of an obvious inflammatory situation, as has been reported previously in some nusinersen-treated patients (apparently without clinical symptomatology). Reference 18 is cited but is an over-simplification of the CSF findings in nusinersen treated patients. They report increased CSF leukocytes in 3 of 72 patients with SMA type 1 treated with nusinersen – infrequent but not rare (despite those authors' characterization).

Response: We appreciate this insightful feedback and have incorporated routine CSF protein level measures into Table 1 as suggested (page 30). Regarding red and white blood cell counts, we observed no apparent inflammation or blood contamination, with all red blood cell counts below 500 cells per microliter and white blood cell counts below 3 cells per microliter in all our cohorts. As a result, we have omitted these details from Table 1. Additionally, we acknowledge the oversimplification of the findings from Reference 18 (now Reference 24) and have expanded the Introduction and Discussion sections of the manuscript, providing a more comprehensive discussion of Reference 18's findings on pages 4, 5, and 10.

6. Results: pro- and anti-apoptotic drivers are mentioned by the authors. Were more specific studies of neuronal apoptosis performed? This has been addressed to a limited degree in earlier literature.

Response: We appreciate the astute observation made by the reviewer regarding the more comprehensive exploration of apoptosis in our study. We have conducted IHC Caspase-3 staining on SMA CNS tissues. This analysis confirmed the presence of apoptosis in the previously identified chromatolytic neurons, evident by the positive staining with the anti-Caspase-3 antibody. We have integrated this IHC data into Figure 3C and mentioned it in the text on page 8 of our manuscript. In addition, we also embarked on a thorough investigation of neuronal apoptosis using the snPATHO-seq data. Specifically, the 'degenerative neurons (DeN)' cluster, as depicted in Figure S4E, unveiled a significant enrichment of GO functions closely associated with neuronal apoptosis, including 'neuron apoptotic process,' 'neuron death,' and 'autophagy.' To provide clarity and visual representation of this enrichment, we have thoughtfully included this relevant information as Figure S4G in our manuscript.

Figure 3C:

Supplementary Figure 4G:

7. Discussion: it would be helpful to address the limitations of linking CSF data from adult SMA patients with neuroimmunopathology in the spinal cord of a single very severe SMA neonate (probably a type 0?). As in point 3 above, your observations and interpretation need to be discussed in some detail in relation to the Nuzzo findings – where are you in alignment with those earlier findings and where are you presenting new and different or contradictory findings, and how these may relate to the SMA type and the age and severity of disease. Do you believe the observations in CSF in adult and less severely affected SMA patients are likely similar to that expected in more severely affected infants? Is your data congruent with other studies in the literature.

Response: We appreciate the valuable insights provided by the reviewer. Indeed, we concur with the idea that the observations in the CSF of adult and less severely affected SMA patients are likely to shed light on the conditions in more severely affected infants. Our study leveraged CSF data from less severely affected SMA adult patients to establish the connection between late-activated cytotoxic CD8 T cells and IL-18-secreting monocytes. This connection serves as compelling evidence for the presence of a higher abundance of GzmB+ CD8+ cells in close proximity to IL-18+ CD68+ cells within the CNS tissues of more severely affected infants (Fig.3D-H).

Nonetheless, it is crucial to acknowledge the limitations in our findings. One significant limitation is our inability to definitively ascertain whether these immune cells primarily reside in the CSF, originate solely within the CNS, migrate from the CNS to peripheral tissues, or emerge from the periphery with the intent of entering the CNS. To address these intricacies, future research may benefit from employing murine SMA models. Such models would provide a robust platform to gain deeper insights into the origin and trafficking dynamics of cytotoxic CD8 T cells and IL-18-secreting monocytes within the context of SMA. This holistic approach could contribute substantially to our understanding of disease pathogenesis. We have discussed this in further detail on page 11 of the manuscript.

Regarding the comparison with the Nuzzo et al.'s findings, it is evident that both our study and theirs depict a consistent trend towards a reduction in inflammatory molecules within the CSF following Nusinersen treatment (as also highlighted in our response to Reviewer #1's second comment). Indeed, there is congruence in our observations. Notably, we examined the baseline and the 6-month post-treatment time points, while Nuzzo et al. conducted their analysis at baseline, 2 months, and 10 months post-treatment. Additionally, Nuzzo et al. monitored CSF

inflammatory molecules across SMA type 1 (n=18), type 2 (n=19), and type 3 (n=11) patients, while our cohort is smaller (n=10), primarily consisting of SMA type 3 patients (n=8).

An intriguing discovery within Nuzzo et al.'s study involving type 3 SMA patients is the significant reduction in IL-1ra levels following a 10-month course of Nusinersen treatment. This stands in contrast to both our findings and theirs, which showed no such reduction at the 2 and 6-month post-treatment intervals. This discrepancy suggests that a specific duration of treatment may be required to effectively ameliorate the elevated IL-1ra levels associated with SMA.

Moreover, our comprehensive analysis, encompassing a 92-plexed panel of molecules, consistently demonstrates a decreasing trend in cytotoxicity-associated proteins, including IL-18R1, CD5, PD-L1, and TNFRSF9, as well as two other inflammatory proteins, CX3CL1 and MMP-1 after 6 months of treatment. It is noteworthy that these molecules were not included in Nuzzo et al.'s 27-plexed panel. Nevertheless, our findings align with the insights derived from our transcriptome data within the CSF, collectively emphasizing the neuro-immunomodulatory impact of Nusinersen therapy.

We have included a detailed discussion of this reference in the discussion section on page 10 of the manuscript.

Reviewer #3

General comments

The manuscript is clearly written and identifies a detailed profile of cell-mediated inflammation that may be contributing to SMA etiology. The N is low, but this is sufficiently acknowledged throughout and consistent with analyses of rare diseases.

Response: We sincerely appreciate the thorough evaluation and valuable insights on our manuscript.

1) The authors focus quite a bit on CD8_2, but they don't describe what CD8_1 is (and from the limited info given, it seems like it might also be cytotoxic). In the later clonal analysis CD8_1 I think has the largest clonotypes which would also be a point of interest. Clearer DEG analysis between CD8_1 and 2 would be helpful (especially because at least at a glance CD8_1 does not appear naïve).

Response: We appreciate the reviewer's insightful comments and have thoroughly addressed the points raised regarding CD8_1 and CD8_2 cells in our study. CD8_1 cells, indeed, possess transcriptional signs of cytotoxic properties, and they do not appear to be in a naive state. To provide a clearer understanding of the distinctions between CD8_1 and CD8_2, we conducted a comprehensive DEG analysis, the results of which are depicted in the below volcano plot, which we integrated into the manuscript as Supplementary Figure 8A and 8B.

Notably, our analysis reveals that CD8_2 cells exhibit a significantly elevated expression of the GNLY gene, which encodes Granulysin, a cytolytic molecule expressed by cytotoxic T cells in the later stages of activation (typically 3–5 days post-activation)^{9,10}. This finding strongly suggests that CD8_2 cells represent a late-activated cytotoxic T cell subset.

Consistent with this observation, our Gene Ontology (GO) enrichment analysis underscores the unique functional profiles of these two subsets. Biological functions related to the later stages of cytotoxicity, such as 'negative regulation of cell killing' and 'negative regulation of leukocyte-mediated cytotoxicity,' are exclusively associated with CD8_2 cells. We have included these data in Supplementary Figure 8 and included an expanded discussion in the Results section on page 8.

Supplementary Figure 8A and 8B:

2) The authors make claims about the abundance of cells between SMA and healthy controls (e.g. line 182), but it's not clear they've provided the right info for this analysis. Rather than just comparing frequencies with the 10X object, they need to compare some total number or absolute frequency of cells back calculated based on the amount of cells that were put into the 10X pipeline and successfully recovered.

Response: We thank the reviewer for this valuable feedback regarding our analysis of cell abundance. In response to the suggestion, we have incorporated an additional comparison of absolute cell counts for all cell clusters in Supplementary Figure 2. While our initial findings indicated a significant increase in cell frequency for both the CD8_2 and NK clusters in the SMA_baseline group. Specifically, we used the total input cell numbers and the relative cluster proportions to calculate this as suggested by the reviewer. We observed a significant increase in the absolute cell count of CD8_2 cells within the SMA_baseline group (Supplementary Figure 2C), suggesting a pronounced association of these cytotoxic CD8_2 cells with the SMA disease in its untreated state. Other cell clusters were not significantly altered when using this approach. It is also worth noting that the absolute cell count of NK cells did not show a statistically significant increase (Supplementary Figure 2D), though there was a discernible trend towards elevation. This contrast may indicate that the impact of cytotoxic CD8_2 cells is more profoundly linked to the SMA disease in its untreated state. We have toned down the importance of NK cells across the manuscript.

Supplementary Figure 2:

3) *A related problem occurs on lines 251—"T cell expansion" is not really what's being measured here, but rather the number of observed clonal expansions. This could have sampling bias as above and should be carefully controlled for based on cells input and recovered. It would be great to have clonal expansions plotted as a chart across samples.*

Response: We appreciate the reviewer's thoughtful consideration of the terminology and data analysis in our study. The reviewer correctly points out that the term "T cell expansion" might not accurately capture the essence of our analysis. Instead, we are primarily assessing the number of observed clonal expansions. We acknowledge the potential for sampling bias in our study and recognize the importance of careful control based on cell input and recovery to ensure the robustness of our results. We discuss this on page 7 of the manuscript.

However, it is essential to note that CSF leukocytes are naturally present in limited numbers, resulting in considerable variability in the TCR information that we could successfully retrieve from patients' CSF T cells. Notably, there is a notably lower quantity of CSF cells in the 10-month-treated patient samples. Given this inherent limitation, performing downsampling to equalize sample sizes becomes infeasible, as it would lead to an undesirable loss of data and compromise the integrity of our analysis. While we cannot entirely eliminate the possibility of sampling bias, we have taken steps to address this concern by comparing the clonal proportions across samples and groups. These comparative data, which provide more insights into our findings, are now presented in Supplementary Figure 7E and 7F (Page 6 of this letter).

4) *The TCR analysis is underutilized. First, figure 4A is somewhat difficult to interpret. Why not use a standard measure of clonality (that accounts for sampling) such as a diversity index like Shannon or Simpson's?*

Response: In response to the reviewer's suggestion, and in line with the comment provided by Reviewer #1 (Comment #5), we have taken specific steps to improve the interpretability of our TCR data. We have now introduced pie plots in Figure 4B (Page 5 of this letter) to visualize the expansion of cells within various T cell populations at different time points following Nusinersen treatment. These pie plots utilize circle radii to represent 'cell counts,' providing a more intuitive and informative depiction of the expanded cell clusters.

Furthermore, we have conducted diversity analysis to evaluate and compare the diversity of clonotypes within our samples. To do this, we have applied the true diversity index, a deliberate choice that is well-suited for datasets like ours, aiming to provide a more accurate representation of diversity. This diversity analysis is detailed in Supplementary Figure 7B (Page 6 of this letter). We believe that this considerably improves the accessibility of our data for the reader.

5) *A second simple TCR analysis would be to compare clonal lineages between time points—are the same cells present in individuals during treatment? Are the expanded clones consistently observed? This would be important info to highlight. This information is available in Supp Table 9 but as it is a sparse table it's difficult to scan to get an impression of the data structure.*

Response: We greatly appreciate the reviewer's insightful feedback and the important questions raised regarding our TCR analysis. In our study, we have examined clonal lineages across

multiple time points in the four patients for whom we collected CSF cells at more than three time points (see Fig.2 for Reviewers only below). Our clonotype tracking analysis has revealed that the same cells are consistently present within individuals during the course of treatment. Moreover, we have observed that the expanded clones, a significant focus of our study, are consistently and reproducibly identified within the same individual over time. It is important to note that these expanded clones are unique to each patient and are not shared between different patients. However, there is one notable exception with a shared clonotype (CSVVDTEAFF; addressed in the following comment).

Figure 2 for Reviewers only: (A) TCR clonal tracking in patient SMA1-1 before treatment (baseline), 6 months (6mo) and 10 months (10mo) after treatment. (B) TCR clonal tracking in patient SMA1-4 before treatment (baseline), 6 months (6mo) and 10 months (10mo) after treatment. (C) TCR clonal tracking in patient SMA1-6 before treatment (baseline), 6 months (6mo) and 10 months (10mo) after treatment. (D) TCR clonal tracking in patient SMA1-7 before treatment (baseline), 2 months (2mo), 6 months (6mo) and 10 months (10mo) after treatment.

6) Last, running some additional (simple) TCR analyses would be informative—first, looking for clusters of related TCR sequences can easily be done by free tools (*tcrdist*, *GLIPH*) or even just applying a simple edit distance. Additionally, trying to match the observed sequences to public

databases might be informative if a match is found. This can be difficult given HLA restrictions and the limited nature of public databases (e.g. vjdb) but as the authors note, a prior study in Alzheimers found expanded CD8 T cells that were able to be mapped to an EBV specific response.

Response: We greatly appreciate the reviewer’s thoughtful suggestions for additional TCR analyses. To address the first point, we have conducted public clonotype analysis, which allowed us to estimate the similarity of samples by considering the number of shared clonotypes. Additionally, we applied hierarchical clustering analysis to assess the distance between samples. The results of these analyses are presented in Supplementary Figure 7C and 6D. Our findings indicate that shared clonotypes are primarily observed within samples collected at different time points from the same patient. However, there is one intriguing exception: the clonotype with the CDR3 sequence - CSVVDTEAFF, which is shared between patients SMA1-1 and SMA1-6. We found no TCR clonotype that is shared among patients and linked to any viral infection upon analysis of public VDJ databases.

Supplementary Figure 7C and 7D:

REFERENCES

1. Krzywinski, M. & Altman, N. Comparing samples—part II. *Nat. Methods* **11**, 355–356 (2014).
2. Hartlehnert, M. *et al.* Bcl6 controls meningeal Th17-B cell interaction in murine neuroinflammation. *Proc. Natl. Acad. Sci. U. S. A.* **118**, e2023174118 (2021).
3. Heming, M. *et al.* Neurological Manifestations of COVID-19 Feature T Cell Exhaustion and Dedifferentiated Monocytes in Cerebrospinal Fluid. *Immunity* **54**, 164-175.e6 (2021).
4. Schafflick, D. *et al.* Integrated single cell analysis of blood and cerebrospinal fluid leukocytes

- in multiple sclerosis. *Nat. Commun.* **11**, 247 (2020).
5. Ostkamp, P. *et al.* A single-cell analysis framework allows for characterization of CSF leukocytes and their tissue of origin in multiple sclerosis. *Sci. Transl. Med.* **14**, eadc9778 (2022).
 6. Touil, H. *et al.* A structured evaluation of cryopreservation in generating single-cell transcriptomes from cerebrospinal fluid. *Cell Rep. Methods* **3**, 100533 (2023).
 7. Roostaei, T. *et al.* Defining the architecture of cerebrospinal fluid cellular communities in neuroinflammatory diseases. <http://biorxiv.org/lookup/doi/10.1101/2021.11.01.466797> (2021) doi:10.1101/2021.11.01.466797.
 8. Bouzid, H. *et al.* Clonal hematopoiesis is associated with protection from Alzheimer's disease. *Nat. Med.* **29**, 1662–1670 (2023).
 9. Jongstra, J. *et al.* The isolation and sequence of a novel gene from a human functional T cell line. *J. Exp. Med.* **165**, 601–614 (1987).
 10. Krensky, A. M. & Clayberger, C. Biology and clinical relevance of granulysin. *Tissue Antigens* **73**, 193–198 (2009).

REVIEWERS' COMMENTS

Reviewer #1 (Remarks to the Author):

The authors have diligently addressed and appropriately incorporated many of the comments raised by the reviewers. However, it is recommended that the authors include the newly generated edits and analyses within the main body of the text. For example, the authors still cite the manuscript #43 as a reference for the 3' scRNA-seq used in this present study although in this PNAS paper, the method is not described in the methods section as claimed by the authors. Another example of the analysis of shared clonotypes between the patients that was requested by Reviewer 3.

In response to the previous critique by Reviewer 1 in Line 228 "granzyme genes are located in close proximity to DeN" The authors reported in Supplementary Figure 4F H&E staining derived from the same tissue section utilized for spatial transcriptomics. The authors state that they "observed blood vessels in close proximity to DeN".

They added that "This spatial arrangement suggests a potential mechanism wherein increased blood vessels in these regions may facilitate the access of cytotoxic cells, proteins, and other relevant substances to neurons".

The increase cytolytic production of granzyme genes in proximity to DeN may be involved in motor neuron degeneration, however, the authors claim that this is due to the infiltration of cytotoxic T cells which is not accurate. Also, Figure S4F is not described in the main text.

Minor:

I suggest adding a graphic summary of the study depicting the study cohort and different types of analyses that was performed with the corresponding n.

Reviewer #3 (Remarks to the Author):

The authors have performed extensive revisions and even obtained additional samples supporting the core findings of the manuscript. I have no further concerns.

Cell-mediated Cytotoxicity within CSF and Brain Parenchyma in Spinal Muscular Atrophy

POINT BY POINT RESPONSE TO REVIEWER#1

Reviewer #1

General comments

The authors have diligently addressed and appropriately incorporated many of the comments raised by the reviewers. However, it is recommended that the authors include the newly generated edits and analyses within the main body of the text. For example, the authors still cite the manuscript #43 as a reference for the 3' scRNA-seq used in this present study although in this PNAS paper, the method is not described in the methods section as claimed by the authors. Another example of the analysis of shared clonotypes between the patients that was requested by Reviewer 3.

In response to the previous critique by Reviewer 1 in Line 228 “granzyme genes are located in close proximity to DeN” The authors reported in Supplementary Figure 4F H&E staining derived from the same tissue section utilized for spatial transcriptomics. The authors state that they “observed blood vessels in close proximity to DeN”. They added that “This spatial arrangement suggests a potential mechanism wherein increased blood vessels in these regions may facilitate the access of cytotoxic cells, proteins, and other relevant substances to neurons”. The increase cytolytic production of granzyme genes in proximity to DeN may be involved in motor neuron degeneration, however, the authors claim that this is due to the infiltration of cytotoxic T cells which is not accurate.

Also, Figure S4F is not described in the main text.

Response: We thank the reviewer for the positive feedback and valuable suggestions. We apologize for any lack of clarity regarding the 3' circ-TCRseq method. It was actually described in supplementary figure S3 of reference #43. To be more explicit: we reported the method first in the paper (<https://doi.org/10.1016/j.immuni.2020.12.011>) now cited as reference #34 in the manuscript. A more detailed description of the method including an extensive schematic are included in publication now cited as reference #56. We make sure to cite both publications.

To ensure readers have comprehensive information on the technique, we have provided more detailed technical steps in supplementary figure S7 of our manuscript. Regarding the analysis of shared clonotypes between patients, we have now incorporated this finding into the Results section on page 8. We agree that we may have overinterpreted the vicinity of cytotoxic T cells to blood vessels as evidence of active infiltration; which is something we cannot prove. We have therefore removed this statement from the manuscript. We added reference to Fig. S4F to the text on page 8.

Minor:

I suggest adding a graphic summary of the study depicting the study cohort and different types of analyses that was performed with the corresponding n.

Response: We greatly appreciate the reviewer's insightful suggestion. In response, we have integrated a graphic summary illustrating the study cohort and the various analyses conducted, complete with corresponding sample numbers. This visual representation can be found in supplementary figure S1A.

Supplementary Figure 1A:

A